# Morpho-Syntactic Deficit in Children with Cochlear Implant: Consequence of Hearing Loss or Concomitant Impairment to the Language System?

**DOI:** 10.3390/ijerph18189475

**Published:** 2021-09-08

**Authors:** Erika Benassi, Sonia Boria, Maria Teresa Berghenti, Michela Camia, Maristella Scorza, Giuseppe Cossu

**Affiliations:** 1Department of Biomedical, Metabolic and Neural Sciences, University of Modena and Reggio Emilia, 42122 Reggio Emilia, Italy; michela.camia@unimore.it (M.C.); maristella.scorza@unimore.it (M.S.); 2Center for the Diagnosis, Treatment and Study of Communication and Socialization Disorders, NPIA, AUSL of Parma, 43125 Parma, Italy; soboria@ausl.pr.it; 3Department of Medicine and Surgery, University of Parma, 43126 Parma, Italy; mariateresa.berghenti@unipr.it; 4Medical Centre of Phoniatrics, 35142 Padova, Italy; gcossu@centrofoniatria.it

**Keywords:** children with hearing loss, cochlear implant, morpho-syntactic impairment, language disorder

## Abstract

Background: Among implanted children with similar duration of auditory deprivation and clinical history, the morpho-syntactic skills remain highly variable, suggesting that other fundamental factors may determine the linguistic outcomes of these children, beyond their auditory recovery. The present study analyzed the morpho-syntactic discrepancies among three children with cochlear implant (CI), with the aim of understanding if morpho-syntactic deficits may be characterized as a domain-specific language disorder. Method: The three children (mean age = 7.2; SD = 0.4) received their CI at 2.7, 3.7, and 5.9 years of age. Their morpho-syntactic skills were evaluated in both comprehension and production and compared with 15 age-matched normal-hearing children (mean age = 6.6; SD = 0.3). Results: Cases 1 and 2 displayed a marked impairment across morphology and syntax, whereas Case 3, the late-implanted child, showed a morpho-syntactic profile well within the normal boundaries. A qualitative analysis showed, in Cases 1 and 2, language deficits similar to those of normal hearing children with Developmental Language Disorder (DLD). Conclusions: We suggest that a severe grammatical deficit may be, in some implanted children, the final outcome of a *concomitant* impairment to the language system. Clinical implications for assessment and intervention are discussed.

## 1. Introduction

In industrialized countries, the vast majority of children with neurosensorial severe to profound deafness can receive one or two cochlear implants (CIs) that can allow them to be exposed to spoken language in natural contexts, offering the opportunity for incidental language learning [1]. The age of implantation is credited to play a crucial role in the language acquisition of the implanted deaf children; intuitively, by making CI available at an early age, one would expect the gap between language development and chronological age to be minimized. A study by Svirsky [2], where 70 children were assessed 4 months before they received their CI and then again at 6, 12, 18, 24, and 30 months after implantation, documented that an earlier CI led to better language proficiency. A study on Italian preschool children [3] revealed a highly significant effect of age at CI surgery on productive vocabulary, mean length of utterance (MLU), and sentence complexity, with preschool females implanted under 1 year showing better performance. While only partial, the restoration of hearing seems to reduce the risk of the child’s missing critical periods of neural development, allowing for a relatively typical maturation of the auditory system [4]. Thus, some authors [5] suggested that implanting a CI in profoundly deaf children at the youngest age provided an optimal opportunity to acquire communication skills that approximated those of their peers with normal hearing. However, despite the CI expanding the possibilities for acoustic recovery in deaf children, improvements in language skills of these children are not always in line with what is expected [1]. Several studies conducted on large populations of implanted children have documented a relevant interindividual variability [1,2,6]. As Svirsky and colleagues [2] pointed out, some children’s language abilities remain severely delayed even after more than 2 years of experience with their cochlear implant. A study conducted on 181 children who received a cochlear implant by age 5 indicated that some 40% of the sample failed to acquire a linguistic proficiency comparable with that of typically developing (TD) children [6]. Nikolopoulos and colleagues [7] indicated that, although language acquisition was faster and more efficient in those children who received a CI before age 4, their acquisition of spoken grammar was considerably delayed. Lund [8] showed that the magnitude of difference in language abilities between children with CI and normally hearing children did not appear related to age of implantation or to duration of implantation, and other studies still found significant differences between children with CI and TD children in many linguistic skills (e.g., semantics, syntax, spoken language, pragmatics) with lower performance in the implanted children [9,10]. For instance, Rinaldi and colleagues [9] selected deaf toddlers that received their CI within the second year of life, experienced binaural stimulation, were not exposed to sign language, had no additional disability, and had parents actively involved in their child rehabilitation; however, despite this careful selection of the sample, they found linguistic skills within normal limits in fewer than half of the children. The study conducted by Wie [11] in children that received bilateral CIs between 5 and 18 months of age found that, after 12–48 months with CI, only 57% of them had expressive language skills within the normative range.

These conflicting effects of the CI on verbal language development have created a significant debate in scientific world, with some researchers claiming that a very early implantation associated to early oral therapy (without exposure to signs) may promote good language performance in children with CI (e.g., [12,13]), whereas other argue that CI and oral therapy alone may not be sufficient to ensure typical language development (e.g., [1,8,14]). In support of this latter view, there is now evidence that environmental factors (e.g., linguistic input, communication modalities used during interactions, and exposure to a sign language) explain a considerably larger proportion of the variance in receptive and productive language than age at implantation (e.g., [15,16,17]). For example, research has provided initial support to the hypothesis that deaf children with CI can develop better spoken language skills when exposed to a sign language [15,16], demonstrating that early exposure to sign language in a bilingual environment allows the child with CI to express ideas and concepts that they were not yet able to vocalize, thus promoting the development of the language system [16]. Furthermore, domain-general neurocognitive processes, such as sequential processing, working memory, and executive functions, may support the language development of these children or, on the contrary, hinder it when impaired (e.g., [18,19,20]).

It is clear that understanding variability in CI outcomes requires a broader perspective that goes beyond hearing alone [1]. Moreover, it is undeniable that, while generally effective in transducing sounds into electrical signals for the brain, CIs remain artificial devices that can only approximate the natural hearing experience. Certainly, the lack of full access to a complete language (verbal and sign language) during the critical perceptual window [21,22] may negatively impact the neurological development of the language areas [22]. Thus, CIs do not necessary protect the child from failures in full language acquisition [14].

As reported in a review by Caselli and colleagues [23], difficulties in comprehension are often greater than difficulties in production, and difficulties in morphosyntactic aspects are greater than those in lexical aspects. Most of the studies that focused on these linguistic abilities investigated English-speaking children with CI. The differences between the English and Italian languages highlight the importance to focus on studies that investigated morpho-syntactic skills in Italian children with CI [24]. Italian has specific linguistic characteristics, with a richer morphology compared to English. In fact, both bound and free morphology (i.e., inflections of nouns and adjectives, verb conjugation, and articles, pronouns, and prepositions) are quite complex [25]. Previous studies [10,26] found that Italian children with CI made more morphological and syntactic errors in their production task, compared with their normal hearing peers, with difficulties in free morphology (such as articles, pronouns, and clitics) in particular. In a recent study, Giustolisi and colleagues [27] investigated the production of third-person singular accusative object clitics (3DO clitics) in 14 Italian-speaking children who received a CI between age 1 and 4. The majority of participants had good language performance; however, some children showed severe difficulties in 3DO clitics tasks. These difficulties could be due to perceptual problems that continue to be present despite the CI [25] and may be influenced by the duration of hearing exposure [26]. However, according to other authors [27,28,29], it is also possible that the language difficulties in some of these children may represent a manifestation of more extensive morpho-syntactic and linguistic impairments.

Hawker and colleagues [29] described six prelingually deaf children with disproportionate morpho-syntactic problems, in both production and comprehension, when they were matched to six other implanted children for etiology, age at implantation, and hearing experience with CI. The authors argued that it is important to consider alternative explanations that locate the source of the problem in the brain, i.e., in the CI or the auditory periphery. It is known that around 4% to 6% of children who are normally hearing have DLD, a condition for which there is a strong genetic predisposition. As reported by Hawker and colleagues [29], “because there is no reason to suppose that deafness protects against DLD, we can anticipate that a similar proportion of children who are deaf will have a genetic predisposition to language impairments […] If a child had the same risk factors and a hearing loss, it seems plausible that this would dramatically impair language acquisition”. In other words, in certain circumstances, the linguistic brain of some implanted children may not be fully equipped for the computation of language and for those children the acquisition of language may follow profoundly divergent patterns [28].

This line of reasoning leads to speculate that some congenitally deaf children may suffer from a concomitant linguistic impairment, which can be further amplified by the sensory loss. The co-occurrence of deafness and linguistic impairment might, to some extent, explain a proportion of the variance in language acquisition among deaf children with CI and their atypical linguistic patterns.

## 2. The Current Study

The current study analyzed the morpho-syntactic discrepancies, in both comprehension and production, among three highly intelligent prelingually deaf children with CI; particular attention was devoted to the production of narrative skills. As the main research question, we wondered if the reported delay in language acquisition among implanted children may be characterized as a domain-specific language disorder. As reported above, children who receive a CI display a large variability in linguistic proficiency, and it is important to remain aware that early implantation does not necessarily ensure that the development of language skills will also fully occur [1]. In the present study, we hypothesized that the morpho-syntactic difficulties of two of three children with CI were not only a consequence of the lack of exposure to a natural and optimal hearing during the critical perceptual window but also indicators of a concomitant impairment to the language system. Specifically, we expected that these two children showed clinical markers similar to those of Italian children with DLD, such as short and few complex sentences, omissions of content and function words, many substitutions of free and bound morphemes, and atypical errors in nominal, adjectival, and verbal inflection [30,31].

To this aim, we suggest that detailed single case studies might positively complement the research conducted on large samples of deaf children. The single-case perspective does indeed provide an ideal observatory for a finer-grained analysis of divergent trajectories in deaf children with CI.

## 3. Materials and Methods

### 3.1. Participants

Three children (mean age = 7.2; SD = 0.4), all females, affected by congenital bilateral profound sensorineural hearing loss (SNHL), participated in this study. We chose these three children because, as described in more detail below, they showed similar clinical, family, and school history, they were operated by the same surgeon and with the same type of CI, they received oral speech therapy by the same speech therapist, they reached the same postimplant auditory threshold, and they had the same nonverbal intelligence level.

All of them were followed at the Otolaryngology and Otoneurosurgery Unit of the University of Parma. All cases fulfilled the following criteria: (1) SNHL identified with universal newborn hearing screening program; (2) no evidence of inner ear malformations at high-resolution CT scan and MRI evaluation; (3) no significant visual, motor, or cognitive problems that might interfere with speech and language development; (4) inclusion in an auditory–verbal (AVT) rehabilitation program; (5) normal-hearing parents and Italian as the native language.

In Case 1, genetic inquiry was negative for mutation of gene Cx 26; in Case 2, genetic inquiry revealed that both child’s parents are healthy bearers of a mutation of the connexin 26 gene (Cx26), thus indicating that her deafness has an inherited cause; in Case 3, genetic inquiry revealed a deletion in Cx 26.

Details concerning age at first amplification, at intervention enrolment, and at CI surgery are presented in Table 1. The click-evoked auditory brainstem response (ABR) was absent at 90 dB nHL in all cases. Unaided preimplant auditory threshold level was ≤90 dB (PTA) for each one of the three deaf children, whereas the aided hearing threshold level was between 50 and 60 dB. From the diagnosis, each child regularly followed a rehabilitation program with a speech therapist.

Case 1 and Case 2 received a CI at 2.7 and 3.7 years of age, respectively, while Case 3 received hers at 5.9 years of age (see Table 1). Although the CI was proposed earlier to this latter family, family personal conditions and concerns about surgery and CI (frequent and understandable among these families) delayed the surgery. All children were implanted with a Nucleus multichannel device (Cochlear Ltd., Sydney, Australia). Postimplant auditory threshold improved markedly, and, about 36 weeks after implantation (and also at the time of our testing), all three children ranged between 25 and 35 dB. Their speech discrimination threshold was assessed by means of the routine clinical Protocol for the Evaluation of Results in Rehabilitative Audiology (PCVRAR) [32]. The findings revealed a high level of accuracy for all of the three children; their scores fell in the highest efficiency category (category 6, i.e., auditory recognition of words in open-set condition), according to [33].

All three children had a high nonverbal intelligence level, as they obtained a score above the 95th percentile on the Raven Colored Progressive Matrices (CPMs) [34] (see Table 1). Table 1 also shows details concerning school and maternal level education for the three cases.

Fifteen age-matched normal-hearing children (mean age = 6.6; SD = 0.3; seven males and eight females), served as a control group for the narrative language task. These TD children were recruited in a primary school of Reggio Emilia (Italy) that voluntarily participated in the study. Children were included if they met the following criteria: (a) they spoke Italian as their first language; (b) they did not have any indications of major cerebral damage, congenital malformations, or visual and hearing impairment; (c) they did not have intellectual disabilities; (d) they received adequate schooling (i.e., regular school attendance). Their cognitive level was within the normal range (for details, see Table 1).

### 3.2. Procedure

The three children with CI were evaluated by their speech therapist at the Otolaryngology and Otoneurosurgery Unit of the University of Parma where the children were followed. Specifically, the deaf girls were examined during the individual assessment conducted in the room in which they received the speech therapy. The Test of Grammatical Comprehension for Children (TCGB), tests for the phonological evaluation of infant speech (PFLI), and CPM tests were carried out in a quiet room and were presented according to test recommendations, in live voice and given orally, during two 45 min sessions.

The TD children were individually met in a quiet room of the school by a psychologist; after a familiarization phase with the examiner, the TCGB, PFLI, and CPM tests were administered during a 45 min session.

Since an Ethics Commission was not present in the University of Parma where and when the study was conducted, the study did not receive formal ethical approval. However, the study was conducted according to the guidelines of the Declaration of Helsinki and to ethical principles by APA and AIP (Italian Psychological Association), as well as in line with current Italian legislation.

The children and their parents were informed in detail about the aims of the study, the voluntary nature of their participation, and their right to withdraw from the study at any time. The children’s parents gave informed written consent for participation in the study, data analysis, and data publication.

### 3.3. Measures

The Test of Grammatical Comprehension for Children (TCGB) [35] TCGB is an Italian test for assessing the comprehension of morphology and syntax. Normative scores are provided for an age range from 3.6 to 8 years of age. The TCGB comprises 76 sentences, aimed at assessing eight different grammatical structures (locative, flexional, active affirmative, active negative, passive affirmative, passive negative, relative sentences, and dative sentences) and a figured album. For each sentence produced by the examiner, the child is required to point to the correct picture among four alternatives. The normative scores are calculated as a function of the number of errors.

Tests for the phonological evaluation of infant speech (PFLI) [36]. PFLI was originally designed as a clinical tool for a qualitative and quantitative analysis of phonological disorder in preschool and school children with language impairment. In clinical practice, the PFLI is widely used for eliciting narrative language. The test comprises 90 brightly colored cartooned pictures of common daily-life actions and situations. For the purposes of our study, we selected 10 pictures (i.e., no. 5, 10, 13, 15, 24, 28, 29, 31, 39, and 48) and used them for prompting a story and thus for examining the morpho-syntactic skills in spontaneous narrative production. Two extra pictures were used for training. Each picture was presented one at a time and the child was required to look carefully and to tell a story about what is happening. No time limitations were set on the child for the description of the picture.

### 3.4. Coding

With regard to the children’s narrative productions (from both the three deaf children and the control group), they were tape-recorded and then transcribed by two blinded examiners (first author and the speech therapist) for scoring and further inspection (see Appendix A). Free and bound morphology was the main target of our linguistic analysis from the children’ elicited stories. We counted the number of occurrences of morphological structures and the number and quality of errors made by each child. We also calculated the MLU (mean length utterance) score for each child, in accordance with the currently agreed criteria [37]. We also considered that the boundaries of the utterance were determined by the occurrence of at least two of the following four criteria [30]: (1) mutual interaction in dialogue; (2) an interval of 300 to 500 m/s of silence taken as a boundary for a sentence; (3) more than one topic from the picture described; (4) a falling prosody, to indicate the conclusion of a sentence. Within each accepted utterance, we counted all the words uttered by the child, including articles, particles, and conjunctions. However, when considering typical expressions, such as, for example, “once upon a time”, the whole utterance was counted as a single word [38]. The variables for scoring the narrative language were quantified according to the following criteria:

*Lexical**diversity.* It represented the number of words produced in the session. This measure reflected both the productivity and the diversity of each child’s language.

*Number of sentences*. The candidate sentences were selected according to the presence of the following categories: noun phrases, verb phrases, and sentence complexity (when, for instance, a relative or subordinate clause was inserted).

*Mean length utterance* (MLU). Utterance length was defined as the mean number of words per utterance. An utterance was defined according to the above criteria.

*Free morphemes*. Free morphemes are meaningless particles that precede the nouns of the sentence. They include articles and particles and, in the Italian language, they specify gender and number in agreement with the noun they refer to.

*Bound morphemes*. Bound morphemes are word parts that modify the meaning of a root morpheme (e.g., the plural masculine “*i*” in the word “bambini” [children], or the first-person plural of the present tense “*amo*” in the word “vediamo” [we see]).

Intercoder agreement between the two examiners was 90%.

### 3.5. Statistical Analysis

All statistical analyses were carried out using SPSS 23.0 for Windows.

With regard to morpho-syntactic comprehension (TCGB), for each deaf child, we calculated raw scores and percentile values for each type of error and for total score, using conversion tables provided by the authors of TCGB [35].

Regarding morpho-syntactic production (PFLI), we conducted statistical comparisons between the three deaf children and TD children considering the following variables: lexical diversity, number of sentences, MLU, and morphological errors (on free and bound morphemes). For each participant and for each variable, we calculated *z*-scores (compared to control group), and the alpha was set at 0.05.

## 4. Results

Results of the qualitative analysis of the deaf children performance in the TCGB are presented in Table 2. The results indicated that mastery of grammatical comprehension was unevenly distributed across the three deaf children; Case 1 and Case 2 displayed a marked impairment across morphology and syntax, whereas Case 3 showed a morpho-syntactic profile well within the normal boundaries. In particular, a qualitative analysis indicated that Case 1, similarly to Case 3, scored no errors in the section of locative sentences, whereas Case 2 had an error score of 3.5. A similar picture emerged from testing relative sentences. However, when we moved to flexional morphology, the situation changed; Case 1 and Case 2 made a very high number of errors (4.5 and 3, respectively), while Case 3 made no error (see Table 2). These numbers, according to the normative data, correspond to 25% and 21% of correct responses for flexional morphology. By considering that, in this domain, the mean percentage of correct responses for their ages equates to 88% and 86%, respectively, we obtained a precise measure of the grammatical distance between these two deaf children (Case 1 and Case 2) and their normal-hearing peers, no less than between them and the other deaf child. Furthermore, Case 2’s language comprehension was by far the more severely impaired, since her responses were inaccurate across all the tested grammatical structures. Indeed, she also made many errors in the dative sentences, which are usually acquired at about 3–4 years of age [31].

The spontaneous language transcriptions are reported in Appendix A. The three implanted children showed a clearcut interindividual discrepancy in their morpho-syntactic production skills; grammatical accuracy and selection of the appropriate lexical items were, in fact, unevenly represented. To provide an example, when presented picture no. 5 of the PFLI (see Appendix A), Case 1 said “*To see the mouse*”, and Case 2 said “*The child watches the mouse … the cheese … the mouse eat* [first person singular verb form] *… cat eats the... the* [singular feminine article] *cheese* [masculine noun]”. Case 3′s description of the same picture, on the contrary, was the following: “*There was a child, whose name was Luca. He put a cheese on the table and there is a cat on the refrigerator*, *whereas there is a mouse on the table and the mouse wants to go and eat the cheese*” (see Appendix A).

The 15 TD children from the control group approached the task with different styles; some of them correctly described the essential characteristics of each picture, others went into picking up any minimal figurative detail, and a few other children provided elaborate invented stories, logically derived from the picture. In particular, these children did so by providing intentions to some of the agents in the picture. This explains the extremely high value of the standard deviation in the number of uttered words (SD = 349.6; see Table 3).

Inspection of the narrative language elicited by the PFLI showed that the discrepancy in lexical production among the three deaf children was remarkable, and that a comparison with the control group’s performance further highlighted these discrepancies. A quantitative analysis indicated a significant difference in the lexical diversity between Case 2 and the control group, while there was no difference between Case 1 and Case 3 and the control group (Table 3). However, whereas Case 1 produced a total of 370 words, Case 3 produced 828 words in describing the same pictures (Table 3).

The number of sentences produced by the three deaf children did not differ from the control group for any of the three deaf children (see Table 3). Instead, the MLU statistically differed between two of the deaf children (Case 1 and Case 2) and the control group, but not between the third child (Case 3) and the TD children (see Table 3).

The lower MLU scores observed in the Case 1 and 2 did not seem just characterized by a reduced number of words per sentence, but also by an abnormal organization of bound morphology. In this linguistic domain, Case 1 and Case 2 made 16 and 12 errors, respectively, whereas Case 3 made four mistakes (Table 3).

Similar conditions were observed with the use of free morphemes; again, Case 1 and Case 2 omitted determiners and prepositions 32 and 35 times, respectively, whilst Case 3 scored just one omission (Table 3). In order to get a representation of these interindividual differences, a comparative description of the picture 15 (see Appendix A) is here reported. Case 1 said, “*The girl plays the* [plural feminine article] *dices* [masculine plural noun]” thus violating gender agreement, in addition to missing the preposition “with” and turning the argument structure from an indirect to a direct one. Case 2 said “*The girl ball … dice … the ball … the table the chair … sat down the girl go* [for goes] *the child teddy bear*”. Case 3′s description was strikingly different: “*There was a girl named Carlotta. She was at the kindergarten and was put to seat. She casts the dice: one is green and the other one is blue … and there is a child named Francesco and he has a teddy bear in his hand and he’s glancing at the wall”*.

## 5. Discussion

Our study investigated three prelingually deaf children who were implanted by 6 years of age and achieved strongly discrepant levels of grammatical skills, despite their similar family and clinical histories, hearing loss levels, postimplant hearing threshold, and nonverbal intelligence levels. One major difference concerned their age at CI implantation, since Case 3 was implanted at 5.9 years of age, while the other two children were implanted at 2.7 and 3.7 years of age. Surprisingly, however, the late-implanted child acquired language skills comparable to those of TD children, whereas the other two children (Cases 1 and 2) showed severe morpho-syntactic deficits, in both comprehension and production.

In the literature, some studies reported that improvement in auditory perception provided by CI has a strong impact on language acquisition of children with hearing loss (e.g., [2,3,4,5]). An early age of implantation, in fact, fosters the recruitment of a vast neuronal network, thus expanding the opportunities for setting up an efficient neural architecture for the acquisition of language. This line of reasoning is supported by measures of P1 latencies, which vary as a function of chronological age and, therefore, provide a measure of the central auditory pathway maturation. A study by Sharma et al. [39] showed that “children with CI who had the longest period of auditory deprivation before implantation demonstrated abnormally long cortical response latencies to speech stimuli. On the contrary, those children who had the shortest period of auditory deprivation (3.5 years or less) demonstrated age-appropriate latency responses” (p. 511). A subsequent study by Sharma et al. [40] further documented that the development of P1 latencies in two congenitally deaf children, who were implanted at 13 and 14 months of age, followed a normal trend. In particular, the authors examined the relationship between P1 latencies and the development of canonical babbling; the findings revealed that the development of P1 response latencies and the development of early communicative behavior followed a similar trajectory. The authors aptly observed that these early stages of speech development in early implanted children “may be positively influenced by the rate of plastic changes in central auditory pathways” [40] (p. 511). In other words, early cochlear implantation appears to be optimal for the development of neural correlates of auditory perception because it allows the children to exploit the development of neuronal connections in the brain.

Yet, as many studies observed (e.g., [6,7,8,9,10,11]), early cochlear implantation does not necessarily correspond to age-appropriate language skills. As reported by Marschark and colleagues [41], it is important to remain aware that CI may not be sufficient to overcome the risk of weaknesses in several aspects of language. Other factors, such as age at first amplification, type of speech therapy, early sign language exposure, maternal educational level, and language input, appear to strongly influence the language development of these children [1]. It should be noted that Cases 1 and 2, relative to Case 3, received the first amplification and therapy a few months later; this condition may have negatively influenced their language acquisition. By contrast, the mothers of the Case 1 and 2 showed higher educational levels than the mother of the Case 3. It is widely documented that the maternal educational level plays an important role in child language acquisition in deaf children with CI [42]. Thus, we can exclude that the linguistic discrepancies between Cases 1–2 and Case 3 are attributable to maternal educational level. There is also recent research that shows that deaf children of hearing parents demonstrate age-level verbal vocabulary growth when exposed to American Sign Language (ASL) by 6 months of age [43]; the lack of access to sign language that characterized the children of the present study may has been a further negative factor for their language development. However, it is important to note that this negative condition affected all three children and not just the first two cases. Thus, taking into account all these conditions and the fact that Cases 1 and 2 showed severely impaired morpho-syntactic abilities, whereas the Case 3 showed age-appropriate skills, it is not unreasonable to put forward the hypothesis that the morpho-syntactic deficit that characterized Cases 1 and 2 may reflect not only the lack of exposure to a natural and optimal hearing during the critical perceptual window, but also some concomitant specific impairment to the language system.

In support of this hypothesis is the qualitatively fine-grained analysis of the language disorders that characterized this single-case investigation and that allowed highlighting atypical morpho-syntactical errors in Cases 1 and 2 quite similar to those made by children with DLD. A glance at the number of errors in the syntactic comprehension task (TCGB) indicates that both more sophisticated and less sophisticated syntactic structures were not properly mastered by both Case 1 and Case 2 (with a mean error of 12.5 and 24.5, respectively), whereas Case 3 had a mean error of 1.5 (well within the normal range). Since the discrepancies between Cases 1–2 and Case 3 cut across the entire set of syntactic structures investigated by the TCGB except one (Case 3 only had one abnormal error rate on passive affirmative sentences, compared to the normative range), it appears that the grammatical errors made by the first two deaf children may reflect an extended impairment to the morpho-syntactic system. Furthermore, the profile of the syntactic errors appeared not to be typical of a younger age. Case 1 and Case 2, for instance, proved to be unable in manipulating the sentences requiring discrimination of flexional morphology; this linguistic competence is clearly within reach of much younger children without difficulty [35]. Our findings are in line with other studies on case reports [28,29,44], thus providing new evidence for the fact that, in certain circumstances, the language acquisition of deaf children with CI may follow profoundly divergent patterns. Our data suggest that, in some cases, we may not be facing a simple morpho-syntactic delay due to perceptual difficulties or to the duration of hearing exposure, but a disorder of the processes of language acquisition.

Inspection of the spontaneous language showed that the discrepancy in morpho-syntactic production among the three deaf children was remarkable and that a comparison with the control group’s performance further highlighted these discrepancies. Furthermore, the qualitative analysis of the spontaneous productions showed, in Cases 1 and 2, atypical morpho-syntactic errors, not only attributable to deafness. As can be seen from the transcripts (see Appendix A), the morpho-syntactic errors of these two children appeared very similar to those observed in Italian normal-hearing children with DLD. In fact, in these two children, it was possible to observe omissions of nouns and verbs, article and prepositions substitutions, atypical verb conjugations, and extremely simple (even telegraphic) sentences, which are all markers of DLD (e.g., [30,31,45]). A small MLU also appears to be an important marker of DLD [46]. Most children with DLD show a language production, measured as MLU, significantly below the age level [47,48]. Hammer and colleagues [49] found that the 75% of the children with DLD scored below age expectations on MLU as opposed to 35.4% in the group of children with CI. Moreover, most of the children with CI achieved appropriate competencies on finite verb production, whereas children with DLD did not. Our findings appear in line with these data on children with DLD; in effect, the MLU was severely impaired in Case 1 and Case 2. These two children reached an MLU of 3.03 and 3.08, respectively, whereas Case 3 and TD children showed an MLU of 8.7 and 8.8, respectively. A more systematic collection of data is clearly required; however, we tentatively suggest that a concomitant occurrence of deafness and morpho-syntactic impairment may explain a proportion of variability in linguistic proficiency of some implanted children.

Children with DLD show that normal hearing per se is not a sufficient condition for language to be acquired at normal rate [50]. There were also studies that found DLD in signing deaf children, supporting the theoretical argument that DLD can be evident regardless of the modality of communication and can affect deaf children [51,52]. For instance, Herman and colleagues [51] looked at narratives in British Sign Language (BSL) and showed that the DLD-diagnosed deaf children performed poorly on verb morphology, and their narratives exhibited less structure and were shorter than those of the TD children. Marshall and colleagues [52] found deaf children with DLD less able to accurately repeat all elements of BSL sentences, including grammatical constructions. These findings seem to indicate that atypicalities in the sign modality show many of the same characteristics as linguistic deficits in hearing children [53].

Our findings appear in line with these data and seem, therefore, to provide new evidence to the hypothesis that other single-case studies have suggested [28,29,44], i.e., that the deafness can, in some cases, be associated with a DLD, as it would seem for Cases 1 and 2. In contrast, the good morpho-syntactic performances observed in Case 3 may be the result of both the earlier amplification and therapy relative to other two children with CI and, in line with our hypothesis, language area integrity.

Limitations of this work should be acknowledged. First, being a single-case study, the generalizability of our findings should be carefully considered. Replication of the present findings with larger samples is clearly needed in the future. In particular, it appears necessary to conduct a detailed qualitative analysis of the language skills in a larger group of deaf children with an earlier age at CI surgery and comparing them with groups of hearing children with DLD and signing deaf children with DLD. Second, the present work did not consider the role of domain-general neurocognitive processes, such as sequential processing, working memory, and executive functions, which several authors found to strongly influence the syntactic abilities of children with DLD [54,55,56], as well as the linguistic outcomes of deaf children with CI [18,19,20]. Third, although we controlled for maternal educational level, we did not investigate other important environmental factors, such as level of parental IQ, maternal language input, and family psychological wellbeing, which several studies have shown to be related to language and learning acquisition in this and other clinical populations [42,57,58]. Future research should move in these directions. Lastly, a further limitation of this study is the absence of neurofunctional correlates, which are crucial if we aim to explain the association between deafness and language system functional deficit. Despite these limitations, the results of this single-case investigation appear sufficient to suggest this association.

## 6. Conclusions

Numerous variables, such as age at first amplification and at beginning language rehabilitation, early sign language exposure, age at implantation, cognitive skills, and maternal education level and language input, may influence the linguistic outcomes of deaf children with CI; as previously reported in this work, these variables have been extensively studied in this clinical population. Instead, few are the studies that investigated if the reported delay in language acquisition among implanted children may be characterized as a domain-specific language disorder. Cases 1 and 2 showed grammar impairments very similar to those of children with DLD. Thus, the profound morpho-syntactic deficits observed in these two cases may be the result of more co-occurring factors: deafness-related factors, such as the lack of full access to a complete language (verbal and sign language) during the critical perceptual window [1,21,22], as well as the presence of a concomitant DLD [28,29,44].

In other words, we are suggesting that the persistent language impairment in some deaf children with CI may be the final outcome of a pre-existing impairment of the language system, combined with the negative effects of a profound congenital deafness. Early identification of domain-specific language impairments in these children may allow for planning and implementation of individualized interventions. To this end, a qualitative and accurate analysis of the type of morpho-syntactic errors appears needed during the language assessment of these children. Furthermore, interventions are needed that specifically act on affected language domains, similarly to that done for children with DLD [59].

## Figures and Tables

**Table 1 ijerph-18-09475-t001:** Clinical data and nonverbal scores from CPMs (Colored Progressive Matrices) for each of the three deaf children and the control group.

	Case 1	Case 2	Case 3	TD Children (*n* = 15)
Chronological age	6.7 years	7.5 years	7.5 years	M = 6.6 (SD = 0.3)
Sex	female	female	female	7 males (47%) and 8 females (53%)
Hearing aids (age at first amplification)	19 months	21 months	13 months	-
Speech therapy (age at initiation)	20 months	21 months	14 months	-
Type of speech therapy	oral	oral	oral	-
Age at CI surgery	2.7 years	3.7 years	5.9 years	-
CPM (raw score; percentile value)	32; >95th	29; >95th	30; >95th	M = 21.1 (SD = 2.70); 90th
Type of school	Traditional school education	Traditional school education	Traditional school education	Traditional school education
Grade of school	grade I	grade I	grade I	grade I
Maternal educational level	High level (high-school diploma)	High level (high-school diploma)	Low level (middle-school graduation)	8 high level (53%), 7 low level (47%)

**Table 2 ijerph-18-09475-t002:** Results (total error raw score, error raw scores for each type of sentence, and percentile values) for the TCGB in the three deaf children with CI.

*TCGB*	Case 1		Case 2		Case 3	
Error Type	Error Raw Score	Percentile Value *	Error Raw Score	Percentile Value *	Error Raw Score	Percentile Value *
Locative sentences	0	>50th	3.5	**<10**th	0	>50th
Flexional sentences	4.5	**<10**th	3	**<10**th	0	>75th
Active affirmative sentences	0.5	**10**th	1.5	**<10**th	0	>50th
Active negative sentences	0.5	50th	3.5	**<10**th	0	>50th
Passive affirmative sentences	1.5	**10**th	3	**<10**th	1	**10**th
Passive negative sentences	4.5	**<10**th	3.5	**<10**th	0.5	25th
Relative sentences	0	>75th	3	**<10**th	0	>50th
Dative sentences	1	**10**th	3.5	**<10**th	0	>25th
Total score	12.5	**10**–**25**th	24.5	**<10**th	1.5	50th

* Percentile values that demonstrate a deficit (defined as ≤10th) or a delay (defined as ≤25th) [35] are in bold.

**Table 3 ijerph-18-09475-t003:** Qualitative analysis of the morpho-syntactic production elicited by PFLI and statistical comparisons between the three deaf children and TD children. Raw scores, *z*-scores and *p*-values of the lexical diversity, number of sentences, MLU, and morphological errors are provided.

		Case 1	Case 2	Case 3	TD Children (*n* = 15)
*PFLI* *		Raw Score (*z*-Score; *p*-Value)	Raw Score (*z*-Score; *p*-Value)	Raw Score (*z*-Score; *p*-Value)	Raw Score (SD)
Lexical diversity		370 (−1.29; 0.09)	262 (−1.60; **0.05**)	828 (0.01; 0.49)	824.2 (349.6)
Number of sentences		122 (0.57; 0.28)	85 (−0.027; 0.46)	95 (0.04; 0.48)	97 (43.2)
MLU		3.03 (SD = 1.6) (−3.20; **<0.001**)	3.08 (SD = 2.1) (−3.17; **<0.001**)	8.7 (SD = 6.6) (−0.05; 0.48)	8.8 (1.8)
Morphological errors		48 (15.3; **<0.001**)	47 (15.03; **<0.001**)	5 (0.55; 0.48)	3.4 (2.9)
	Free	32	35	1	1.9 (2.5)
	Bound	16	12	4	1.5 (1.5)

* Tests for the phonological evaluation of infantile speech (PFLI) [36]. PFLI was used for eliciting narrative language. Significant results are in bold.

## Data Availability

The data presented in this study are available on request from the corresponding author.

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
