# Peer review of "Morpho-Syntactic Deficit in Children with Cochlear Implant: Consequence of Hearing Loss or Concomitant Impairment to the Language System?"

_ijerph, 2021, doi:10.3390/ijerph18189475_

Round 1

Reviewer 1 Report

Benassi E et al., have presented a case study in support of their hypothesis that language development impairment in deaf children cannot be fully accounted for by auditory perceptual difficulties but may be due to concomitant damage to the linguistic system. My main concern is that the results of the study are being generalized even though the conclusions are being drawn from a very small sample size (n=3). The authors have raised this limitation in the discussion. Despite this limitation, in my opinion the findings are interesting and could be of interest to other researchers in the field.

Minor comment: Throughout the manuscript there are paragraphs that have only one or two sentences. The authors can look into merging these sentences with the above or below paragraphs wherever appropriate.

Author Response

We thank Reviewer 1 for having recognised the research value of this investigation.

Following the suggestion of the Reviewer 1, we looked into merging some paragraphs.

Reviewer 2 Report

I appreciated your work, even if the sample of your patients is small, and the high quality of the references.

Few minor reviews are suggested:

  • how many weeks after implantation did the patients reach the hearing target?
  • " they did not have any indications of major cerebral damage, 211
    congenital malformations, or neurological, visual or hearing impairment " (Line 211) : how did you managed to exclude neurological diseases?
  • a mean level of maternal education is missing in TD group
  • How did you used a PFLI test on subjects older than 5 years old? What normative data did you used?
  • I doubt that all tests were administrated in one 45 minute session, you should better specify how many session did you attended

Author Response

We thanks Reviewer 2 for having appreciated our work and for these comments and suggestions.

  • how many weeks after implantation did the patients reach the hearing target?

R: We added this information in the manuscript (see Materials and Methods, Par. Participants)

"Postimplant auditory threshold improved markedly, and about 36 weeks after implantation (and also at the time of our testing) all three children ranged between 25 and 35 dB."

  • they did not have any indications of major cerebral damage, 211
    congenital malformations, or neurological, visual or hearing impairment " (Line 211) : how did you managed to exclude neurological diseases?

R: We have corrected (see Materials and Methods, Par. Participants)

"(b) they did not have any indications of major cerebral damage, congenital malformations, visual and hearing impairment;"

  • a mean level of maternal education is missing in TD group

R: We added this datum in the Table 1 (see Table 1)

  • How did you used a PFLI test on subjects older than 5 years old? What normative data did you used?

R: We used PFLI test for eliciting narrative production in our sample; as described in the manuscript, we did not use normative data but we compared the morpho-syntactic production and errors of the three children with CI with those of the fifteen TD children (see Materials and Methods, Par. Statistical analysis). 

  • I doubt that all tests were administrated in one 45 minute session, you should better specify how many session did you attended

R: The Reviewer 2 is correct. We corrected this information in the manuscript (see Materials and Methods, Par. Procedure)

"Test of Grammatical Comprehension for Children (TCGB), Tests for the phonological evaluation of infant speech (PFLI) and CPM tests were carried out in a quiet room and were presented according to test recommendations, in live voice and given orally, during two 45-min sessions."

This manuscript is a resubmission of an earlier submission. The following is a list of the peer review reports and author responses from that submission.

Round 1

Reviewer 1 Report

The authors present a case study in support of their hypothesis that language development impairment in deaf children cannot be fully accounted for by auditory perceptual difficulties but may be due to concomitant damage to the linguistic system. The conclusions presented by the authors seem to agree with the results of the case study. However, as the authors have aptly acknowledged in the discussion more research using larger sample size and control groups is required before the findings of this case study can be generalized. Despite the limitations, the findings may be of interest to others researchers in the field.

Author Response

Comments and Suggestions for Authors

The authors present a case study in support of their hypothesis that language development impairment in deaf children cannot be fully accounted for by auditory perceptual difficulties but may be due to concomitant damage to the linguistic system. The conclusions presented by the authors seem to agree with the results of the case study. However, as the authors have aptly acknowledged in the discussion more research using larger sample size and control groups is required before the findings of this case study can be generalized. Despite the limitations, the findings may be of interest to others researchers in the field.

R: We thank Reviewer for having recognized the research value and strengths of this investigation.

Reviewer 2 Report

behavsci-1267580

Morpho-syntactic impairment in highly intelligent children with cochlear implant: consequence of hearing loss or concomitant damage to the linguistic system?

This statement is extremely strong to open your paper; “The advent of cochlear implants (CIs) has enormously improved the quality of sensory perception in children with hearing loss, thus providing optimal conditions for subsequent language acquisition” Your citations that you use to support this statement are not sufficient –two of the three are from the 1990’s. You later add some more recent citations but the review of those that show positive versus negative findings are not sufficient in this section. There is a strong controversy regarding this epistemology and you do not even acknowledge that issue. Moog and Geers’ paper that stated that Cis should not sign as well created a significant controversy and was responded to by a large group of well know researchers. At the least these issues need to be included in your paper.

There is research from DCAL and David Qunitos Pozos on Deaf children in Deaf families who seem to have specific language impairment—none of this is included in your paper.  In addition, those children would be better controls for your study. Your argument that linguistic impairments and sensory impairments have neurological basis is one that I cannot accept as claimed.

Rachel Mayberry and Pat Kuhl’s research suggests that language exposure during the perceptual window (babbling rom 6 months to 11 months) impacts the neurological development of the language areas regardless of the modality of the input. Your participants did not have full access to a complete language during this period which I suggest is a better theory to explain your data.

At line 129 you suggest that detailed single case studies would be beneficial.  Your case that has a CI and shows almost typical language acquisition and development is a logical choice and you do not mention that in your discussion.

Participants and their families are not described in enough detail. There are so many confounding variables with these three cases, generalization is impossible and you seem to go beyond your data in the discussion; note the second sentence. Your control participants are from Reggio Emilia so I have to assume that they are using that curriculum, which is one that is more child centered rather than the AVT that your 3 cases are described as experiencing. Moreover, there is no discussion of family advocacy, education, or income—all critical variables in child development. You have a large variability in age of CI implant that makes it difficult to understand the background of your CI participants---why was one implanted after age 5? There is no discussion of this idea; this level of description does not allow replication of your study.

The procedure would be described in more detail.

Statistics should be clearer---the percentiles are not clear in Table 2. In Table 3 why don’t the TD participants have z scores and p values?

Discussion

The extraordinary improvement in auditory perception provided by CI has a dramatic impact on language acquisition of children with hearing loss.

The above statement is not supported with citations or even with your data.  This statement seems to be going beyond your data. Your comparisons with CI and Hearing Aids again reflects a limited view of the most improvement for deaf children in language acquisition.  The focus only on auditory perception does not demonstrates effective language, cognitive, or social emotional development for deaf children. Again, this philosophy is shown in your discussion of Sharma’s work. Mayberry and Kuhn show that neurology is impacted by a lack of access (full access) to language. There is research that shows that with access to a visual language by 6 months vocabulary development is equivalent to hearing children (Casselli et al, 2021—from Boston University) even if their parents are hearing.

Our data suggest that, in some cases, we may not be facing a simple morpho-syntactic 433 delay due to perceptual difficulties or to the duration of hearing exposure, but a qualitative distortion of the process of language acquisition. A more systematic collection of data is clearly required, yet we feel to suggest that a concomitant occurrence of deafness and 436 morpho-syntactic impairment may explain the variability in linguistic proficiency of  many implanted children.  

The statements above from your discussion do not take into account the large body of research that looks at neurological development of language acquisition in deaf children with full access to language. I wonder about your case 3 and why they are different; what is your hypothesis. I do not find your warrants strong enough to support either your results or discussion.

Without neurological studies and evidence I cannot accept your conclusion. Additionally, you need to compare unsuccessful CI individuals with Deaf children who should develop a sign language but do not prior to making this leap in your hypothesis development.

In terms of language there are a few issues but mostly the English is accurate.

Author Response

R: We thank the Reviewers  for the interest in our study, for the time spent on reviewing our manuscript and for the precious comments and suggestions

Comments and Suggestions for Authors

This statement is extremely strong to open your paper; “The advent of cochlear implants (CIs) has enormously improved the quality of sensory perception in children with hearing loss, thus providing optimal conditions for subsequent language acquisition” Your citations that you use to support this statement are not sufficient –two of the three are from the 1990’s. You later add some more recent citations but the review of those that show positive versus negative findings are not sufficient in this section. There is a strong controversy regarding this epistemology and you do not even acknowledge that issue. Moog and Geers’ paper that stated that Cis should not sign as well created a significant controversy and was responded to by a large group of well know researchers. At the least these issues need to be included in your paper.

R: We thank the Reviewer 2 for these comments and suggestions that allow us to better explain our hypothesis, the uniqueness of our study and its important contribution to this research landscape. Following these suggestions, we modified some sections of the Introduction, adding recent literature that shows post-CI conflicting results in deaf children and from which the strong controversy regarding this epistemology emerges. We also added some studies that highlight the important contribution that an early full access to language (including sign language) may have on verbal language acquisition in these children (see Introduction, pp. 2-3 and see Discussion, pp. 14-15).

“In industrialized countries, the vast majority of children with neurosensorial severe to profound deafness can receive one or two cochlear implants (CIs) that can allow them to be exposed to spoken language in natural contexts, offering the opportunity for incidental language learning [1]. The age of implantation is credited to be playing a crucial role in the language acquisition of the implanted deaf children, intuitively, by making CI available at an early age, one would expect the gap between language development and chronological age to be minimized. A study by Svirsky [2], where 70 children were assessed 4 months before they received their CI and then again at 6, 12, 18, 24 and 30 months after implantation, documented that the earlier the CI, the better the language proficiency. A study on Italian pre-school children [3] revealed a highly significant effect of age at CI surgery on productive vocabulary, mean length of utterance (MLU) and sentence complexity, with pre-school females implanted under 1 year showing better performances. While only partially, the restoration of hearing seems to reduce the risk of the child’s missing critical periods of neural development, allowing for a relatively typical maturation of the auditory system [4]. Thus, some authors [5] suggested that implanting a CI in profoundly deaf children at the youngest age provided an optimal opportunity to acquire communication skills that approximated those of their peers with normal hearing.

However, despite the CI is expanding the possibilities for acoustic recovery in deaf children, improvements in language skills of these children are not always in line with what is expected [1].

Several studies conducted on large populations of implanted children have documented a relevant inter-individual variability [1-2, 6]. As Svirsky and colleagues [2] pointed out, some children’s language abilities remain severely delayed even after more than two years of experience with their cochlear implant. A study conducted on 181 children who received a cochlear implant by age 5 indicated that some 40% of the sample failed to acquire a linguistic proficiency comparable with that of typically developing (TD) children [6]. Nikolopoulos and colleagues [7] indicated that, although language acquisition was faster and more efficient in those children who received a CI before age 4, yet their acquisition of spoken grammar was considerably delayed. Lund [8] showed that the magnitude of difference in language abilities between children with CI and normally hearing children did not appear related to age of implantation or to duration of implantation, and still other studies found significant differences between children with CI and TD children in many linguistic skills (e.g., semantics, syntax, spoken language, pragmatics) with lower performances in the implanted children [9-10]. For instance, Rinaldi and colleagues [9] selected deaf toddlers that received their CI within the second year of life, experienced binaural stimulation, were not exposed to sign language, had not additional disability and had parents actively involved in their child rehabilitation; but despite this careful selection of the sample, they found linguistic skills within normal limits in less than half of the children. The study conducted by Wie [11] in children that received bilateral CIs between 5 and 18 months of age found that, after 12-48 months with CI, only 57% of them had expressive language skills within the normative range.

These conflicting effects of the CI on verbal language development have created a significant debate in scientific world, with some researchers that claim that a very early implantation associated to early oral therapy (without exposure to signs) may promote good language performances in children with CI [e.g., 12-13], whereas other argue that CI and oral therapy alone may not be sufficient to ensure typical language development [e.g., 1, 8, 14]. In support of this latter view, there is now evidence that environmental factors (e.g., linguistic input, communication modalities used during interactions, exposure to a sign language) explain a considerably larger proportion of the variance in receptive and productive language than age at implantation [e.g., 15-17]. For example, research has provided initial support to the hypothesis that deaf children with CI can develop better spoken language skills when exposed to a sign language [15-16], demonstrating that early exposure to sign language in a bilingual environment allow the child with CI to express ideas and concepts that he/she was not yet able to speak, thus promoting the development of the language system [16]. Also, domain-general neurocognitive processes, such as sequential processing, working memory, executive functions, may support the language development of these children, or on the contrary hinder it when impaired [e.g., 18-20].

It is clear that understanding variability in CI outcomes requires a broader perspective that goes beyond hearing alone [1]. Also, it is undeniable that, while generally effective in transducing sounds into electrical signals for the brain, CIs remain artificial devices that can only approximate the natural hearing experience. Certainly, the lack of full access to a complete language (verbal and sign language) during the critical perceptual window [21-22] may negatively impact on the neurological development of the language areas [22]. Thus, CIs do not necessary protect the child from failures in full language acquisition [14].”

“In other words, early cochlear implantation appears to be optimal for the development of neural correlates of auditory perception because it allows the children to exploit the development of neuronal connections in the brain.

Yet, as many studies observed [e.g., 6-11], early cochlear implantation does not necessarily correspond to age-appropriate language skills. It is important to remain aware that CI may not be sufficient to overcome the risk of weaknesses in several aspects of language [41].

Other factors, such as age at first amplification, type of speech therapy, early sign language exposure, maternal educational level and language input, appear to strongly influence the language development of these children [1].”

There is research from DCAL and David Qunitos Pozos on Deaf children in Deaf families who seem to have specific language impairment—none of this is included in your paper.  In addition, those children would be better controls for your study. Your argument that linguistic impairments and sensory impairments have neurological basis is one that I cannot accept as claimed.

R: We thank the Reviewer 2 for these suggestions that allow us to provide further literature to support our hypothesis of a possible co-occurrence of hearing loss effects and language impairment in two children of our study. We better explained our hypothesis in the Introduction and we added this literature concerning deaf children who seem to have SLI (today better defined as Developmental Language Disorder - DLD) in the Discussion (see Introduction, p. 4; see The current study, p. 4-5; see Discussion, pp. 16-17).

“Hawker and colleagues [29] described six prelingually deaf children with disproportionate morfo-syntactic problems, in both production and comprehension, when they were matched to other six implanted children for etiology, age at implantation, and hearing experience with CI. The authors argued that it is important to consider alternative explanations that locate the source of the problem in the brain, that in CI or the auditory periphery. It is known that around 4% to 6% of children who are normally hearing have DLD, a condition for which there is a strong genetic predisposition. As reported by Hawker and colleagues [29], “because there is no reason to suppose that deafness protects against DLD, we can anticipate that a similar proportion of children who are deaf will have a genetic predisposition to language impairments […] If a child had the same risk factors and a hearing loss, it seems plausible that this would dramatically impair language acquisition”. In other words, in certain circumstances, the linguistic brain of some implanted children may not be fully equipped for the computation of language and for those children the acquisition of language may follow profoundly divergent patterns [28].

This line of reasoning leads to speculate that some congenitally deaf child may suffer from a concomitant linguistic impairment, which can be further amplified by the sensory loss. The co-occurrence of deafness and linguistic impairment might, to some extent, explain a proportion of the variance in language acquisition among deaf children with CI and their atypical linguistic patterns.”

“As reported above, children who receive a CI display a large variability in linguistic proficiency and it is important to remain aware that the early implantation does not necessarily ensure that the development of language skills will also fully occur [1]. In the present study, we hypothesized that the morpho-syntactic difficulties of two of three children with CI were not only consequence of the lack of exposure to a natural and optimal hearing during the perceptual window but also indicators of a concomitant impairment to the language system.

Specifically, we expected that these two children showed clinical markers similar to those of Italian children with DLD, such as short and few complex sentences, omissions of content and function words, many substitutions of free and bound morphemes, and atypical errors in nominal, adjectival and verbal inflection [30-31].”

“Children with DLD show that normal hearing per se is not a sufficient condition for language to be acquired at normal rate [50]. There were also studies that found DLD in signing deaf children, supporting the theoretical argument that DLD can be evident regardless of the modality in which is communicated and can affects deaf children [51-52]. For instance, Herman and colleagues [51] looked at narratives in British Sign Language (BSL) and showed that the DLD-diagnosed deaf children performed poorly on verb morphology, and their narratives exhibited less structure and were shorter than those of the TD children. Marshall and colleagues [52] found deaf children with DLD less able to accurately repeat all elements of BSL sentences, including grammatical constructions. These findings seem to indicate that atypicalities in the sign modality show many of the same characteristics as linguistic deficits in hearing children [53].

Our findings appear in line with these data and seem therefore to provide new evidence to the hypothesis that other single case studies have already suggested [28-29, 44], namely that the deafness could in some case be associated with a DLD, as it would seem in the Case 1 and 2. By contrast, the good morpho-syntactic performances observed in the Case 3 may be the result of both the earlier amplification and therapy relative to other two children with CI and, in line with our hypothesis, language areas integrity.”

Rachel Mayberry and Pat Kuhl’s research suggests that language exposure during the perceptual window (babbling rom 6 months to 11 months) impacts the neurological development of the language areas regardless of the modality of the input. Your participants did not have full access to a complete language during this period which I suggest is a better theory to explain your data.

R: We thank the Reviewer 2 for this interesting suggestion. We modified the Introduction adding these studies and we hypothesized that the profound morpho-syntactic deficits in some deaf children with CI (as the Case 1 and 2) may be the result of more co-occurring factors, including the lack of full access to a complete language (verbal and sign language) during the perceptual window and also the presence of a concomitant damage to the linguistic system (DLD) (see Introduction, p. 3; see The current study, p. 5; see Discussion, p.15; see Conclusions, p. 17-18).

“These conflicting effects of the CI on verbal language development have created a significant debate in scientific world, with some researchers that claim that a very early implantation associated to early oral therapy (without exposure to signs) may promote good language performances in children with CI [e.g., 12-13], whereas other argue that CI and oral therapy alone may not be sufficient to ensure typical language development [e.g., 1, 8, 14]. In support of this latter view, there is now evidence that environmental factors (e.g., linguistic input, communication modalities used during interactions, exposure to a sign language) explain a considerably larger proportion of the variance in receptive and productive language than age at implantation [e.g., 15-17]. For example, research has provided initial support to the hypothesis that deaf children with CI can develop better spoken language skills when exposed to a sign language [15-16], demonstrating that early exposure to sign language in a bilingual environment allow the child with CI to express ideas and concepts that he/she was not yet able to speak, thus promoting the development of the language system [16]. Also, domain-general neurocognitive processes, such as sequential processing, working memory, executive functions, may support the language development of these children, or on the contrary hinder it when impaired [e.g., 18-20].

It is clear that understanding variability in CI outcomes requires a broader perspective that goes beyond hearing alone [1]. Also, it is undeniable that, while generally effective in transducing sounds into electrical signals for the brain, CIs remain artificial devices that can only approximate the natural hearing experience. Certainly, the lack of full access to a complete language (verbal and sign language) during the critical perceptual window [21-22] may negatively impact on the neurological development of the language areas [22]. Thus, CIs do not necessary protect the child from failures in full language acquisition [14].”

“In the present study, we hypothesized that the morpho-syntactic difficulties of two of three children with CI were not only consequence of the lack of exposure to a natural and optimal hearing during the perceptual window but also indicators of a concomitant impairment to the language system.”

“Other factors, such as age at first amplification, type of speech therapy, early sign language exposure, maternal educational level and language input, appear to strongly influence the language development of these children [1]. It should be noted that the Case 1 and 2, relative to Case 3, received the first amplification and therapy a few months later; this condition may have negatively influence their language acquisitions. By contrast, the mothers of the Case 1 and 2 showed higher educational levels than mother of the Case 3. It is widely documented that the maternal educational level plays an important role on child language acquisition in deaf children with CI [42]. Thus, we can exclude that the linguistic discrepancies between Cases 1-2 and the Case 3 are attributable to maternal educational level. There is also recent research that shows that deaf children of hearing parents demonstrate age-level verbal vocabulary growth when exposed to American Sign Language (ASL) by 6 months of age [43]; the lack of access to sign language that characterized the children of the present study may has been a further negative factor for their language development. However, it is important to note that this negative condition affected all three children and not just the first two cases. Thus, taking into account all these conditions and the fact that Cases 1 and 2 showed severely impaired morpho-syntactic abilities whereas the Case 3 age-appropriate skills, it is not unreasonable to put forward the hypothesis that the morpho-syntactic deficit that characterized the Case 1 and 2 may reflects not only the lack of exposure to a natural and optimal hearing during the critical perceptual window but also some concomitant specific impairment to the language system.

In support of this hypothesis, is the qualitatively fine-grained analysis of the language disorders that characterized this single case investigation and that allowed to highlight atypical morpho-syntactical errors in Case 1 and 2 quite similar to those made by children with DLD.”

“Numerous variables, such as age at first amplification and at beginning language rehabilitation, early sign language exposure, age at implantation, cognitive skills, and maternal education level and language input, may influence the linguistic outcomes of deaf children with CI; as previously reported in this work, these variables have been extensively studied in this clinical population. Instead, few are the studies that investigated if the reported delay in language acquisition among implanted children may be characterized as a domain specific language disorder.Case 1 and 2  showed grammar impairments very similar to those of children with DLD. Thus, the profound morpho-syntactic deficits observed in these two cases may be the result of more co-occurring factors: deafness-related factors, such as the lack of full access to a complete language (verbal and sign language) during the critical perceptual window [1, 21-22], and also the presence of a concomitant DLD [28-29, 44].

In other words, we are suggesting that the persistent language impairment in some deaf children with CI may be the final outcome of a pre-existing impairment to the language system, combined with the negative effects of a profound congenital deafness.”

At line 129 you suggest that detailed single case studies would be beneficial.  Your case that has a CI and shows almost typical language acquisition and development is a logical choice and you do not mention that in your discussion.

R: Following the suggestion of the Reviewer 2, we integrated the Discussion with some considerations regarding the Case 3 that showed normal linguistic abilities (see Discussion, pp. 14-15).

“One major difference concerned their age at CI implantation, since Case 3 was implanted at 5.9 years of age, while the other two children at 2.7 and 3.7 years of age, respectively. Surprisingly, however, the late-implanted child acquired language skills comparable to those of TD children, whereas the other two children (Cases 1 and 2) showed severe morpho-syntactic deficits, in both comprehension and production.”

“It should be noted that the Case 1 and 2, relative to Case 3, received the first amplification and therapy a few months later; this condition may have negatively influence their language acquisitions. By contrast, the mothers of the Case 1 and 2 showed higher educational levels than mother of the Case 3. It is widely documented that the maternal educational level plays an important role on child language acquisition in deaf children with CI [42]. Thus, we can exclude that the linguistic discrepancies between Cases 1-2 and the Case 3 are attributable to maternal educational level. There is also recent research that shows that deaf children of hearing parents demonstrate age-level verbal vocabulary growth when exposed to American Sign Language (ASL) by 6 months of age [43]; the lack of access to sign language that characterized the children of the present study may has been a further negative factor for their language development. However, it is important to note that this negative condition affected all three children and not just the first two cases. Thus, taking into account all these conditions and the fact that Cases 1 and 2 showed severely impaired morpho-syntactic abilities whereas the Case 3 age-appropriate skills, it is not unreasonable to put forward the hypothesis that the morpho-syntactic deficit that characterized the Case 1 and 2 may reflects not only the lack of exposure to a natural and optimal hearing during the critical perceptual window but also some concomitant specific impairment to the language system.

In support of this hypothesis, is the qualitatively fine-grained analysis of the language disorders that characterized this single case investigation and that allowed to highlight atypical morpho-syntactical errors in Case 1 and 2 quite similar to those made by children with DLD.”

Participants and their families are not described in enough detail. There are so many confounding variables with these three cases, generalization is impossible and you seem to go beyond your data in the discussion; note the second sentence. Your control participants are from Reggio Emilia so I have to assume that they are using that curriculum, which is one that is more child centered rather than the AVT that your 3 cases are described as experiencing. Moreover, there is no discussion of family advocacy, education, or income—all critical variables in child development. You have a large variability in age of CI implant that makes it difficult to understand the background of your CI participants---why was one implanted after age 5? There is no discussion of this idea; this level of description does not allow replication of your study.

R: With regard to the first issue raised by the Reviewer 2 (that is, exposure to a curriculum more child centered in TD group rather than the AVT experienced by the three children with CI), we think that the three children with CI and the TD children are comparable for this condition, because the three cases attended the same kind of school of the TD children, namely schools in Reggio Emilia and in Emilia Romagna region that show the same child centered curriculum (see “Type of school” in Table 1). We agree with the Reviewer 2 that some familiar variables, education level in particular, are very important in child language development in both TD children and children with CI; following this suggestion, we added this information in Table 1 (see Table 1); we also added some considerations and literature regarding this issue in the manuscript (see Discussion, p. 15, 17). With regard to the age of CI, following the suggestion of the Reviewer 2, we described why Case 3 was implanted after age 5 in the Participant section (see Method, par. Participants, p. 5). We believe that now the study provides sufficient information about the participants to allow its replication.

“By contrast, the mothers of the Case 1 and 2 showed higher educational levels than mother of the Case 3. It is widely documented that the maternal educational level plays an important role on child language acquisition in deaf children with CI [42]. Thus, we can exclude that the linguistic discrepancies between Cases 1-2 and the Case 3 are attributable to maternal educational level.”

“Limitations of this work should be acknowledged. First, being a single case study, the generalizability of our findings should be carefully considered. Replication of the present findings with larger samples is clearly needed in the future. In particular, it appears necessary to conduct a detailed qualitative analysis of the language skills in a larger group of deaf children with an earlier age at CI surgery and comparing them with both a group of hearing children with DLD and signing deaf children with DLD. Second, the present work did not consider the role of domain-general neurocognitive processes, such as sequential processing, working memory, executive functions that several authors found to strongly influence the syntactic abilities of children with DLD [54-56] and also the linguistic outcomes of deaf children with CI [18-20]. Third, although we controlled for maternal educational level, we did not investigate other important environmental factors, such as level of parental IQ and the maternal language input, that several studies have shown to explain a considerably proportion of the variance in language acquisition [42]. In effect, it would have been important to check whether Case 1 and 2 received the same kind of language exposure than Case 3 at the early age. Future research should move in these directions.”

“Case 1 and Case 2 received a CI at 2.7 and 3.7 years of age, respectively, while Case 3 at 5.9 years of age (see Table 1). Although the CI was proposed earlier to this latter family, family personal conditions and concerns about surgery and CI (frequent and understandable among these families) delayed the surgery.”

The procedure would be described in more detail.

 R: Following this suggestion, the procedure was described in more detail (see Method, par. Procedure, pp. 8-9).  

“The three children with CI were evaluated by their speech therapist at the Otolaryngology and Otoneurosurgery Unit of the University of Parma where the children were followed. Specifically, the deaf girls were examined during the individual assessment conducted in the room in which they received the speech therapy. TCGB, PFLI and CPM tests were carried out in a quiet room and were presented according to test recommendations, in live voice and given orally, during a 45-min session.

The TD children were individually met in a quiet room of the school by a psychologist; after a familiarization phase with the examiner, the TCGB, PFLI and CPM tests were administered during a 45-min session.

The study met ethical guidelines for human subject protections (Declaration of Helsinki). The children and their parents were informed in detail about the aims of the study, the voluntary nature of their participation, and their right to withdraw from the study at any time. The children’s parents gave informed written consent for participation in the study, data analysis, and data publication.”

Statistics should be clearer---the percentiles are not clear in Table 2. In Table 3 why don’t the TD participants have z scores and p values?

R: We modified statistics and percentiles value in Table 2 making them clearer (see Method, par. Statistical analysis, p. 9; see Table 2).  In Table 3, TD participants do not have z scores and p values because these two values refer to how much the performance of each child with CI statistically differs from the TD group and not vice versa (and this is statistically correct, while the opposite would not be; for detail regarding this statistical procedure see Field, 2009, pp. 26, 134-138; for similar statistical analyses in a similar single case study, see also De Stefano et al., 2019).

“All statistical analyses were carried out using SPSS 23.0 for Windows.

With regard to morpho-syntactic comprehension (TCGB), for each deaf child, we calculated raw scores and percentile values for each type of error and for total score, using conversion tables provided by the authors of TCGB [35].

Regarding morpho-syntactic production (PFLI), we conducted statistical comparisons between the three deaf children and TD children on the following variables: lexical diversity, number of sentences, MLU and morphological errors (on free and bound morphemes). For each participant and for each variable we calculated z scores (compared to control group) and the alpha was set at 0.05.”

Discussion

The extraordinary improvement in auditory perception provided by CI has a dramatic impact on language acquisition of children with hearing loss.

The above statement is not supported with citations or even with your data.  This statement seems to be going beyond your data. Your comparisons with CI and Hearing Aids again reflects a limited view of the most improvement for deaf children in language acquisition.  The focus only on auditory perception does not demonstrates effective language, cognitive, or social emotional development for deaf children. Again, this philosophy is shown in your discussion of Sharma’s work. Mayberry and Kuhn show that neurology is impacted by a lack of access (full access) to language. There is research that shows that with access to a visual language by 6 months vocabulary development is equivalent to hearing children (Casselli et al, 2021—from Boston University) even if their parents are hearing.

R: Following the comments and suggestions of the Reviewer 2, we modified the Discussion, focusing not only on auditory perceptual but also on other variables that could explain the improvement or, on the contrary, delays for deaf children in language acquisition. We thank the Reviewer 2 for having suggested this literature that we added in the manuscript and allows us to better discuss our results (see Discussion, pp. 14-15).

“Our study investigated three prelingually deaf children who were implanted by 6 years of age and achieved strongly discrepant levels of grammatical skills, despite their similar family and clinical histories, hearing loss levels, postimplant hearing threshold, and non-verbal intelligence levels.

One major difference concerned their age at CI implantation, since Case 3 was implanted at 5.9 years of age, while the other two children at 2.7 and 3.7 years of age, respectively. Surprisingly, however, the late-implanted child acquired language skills comparable to those of TD children, whereas the other two children (Cases 1 and 2) showed severe morpho-syntactic deficits, in both comprehension and production.

In literature, some studies reported that improvement in auditory perception provided by CI has a strong impact on language acquisition of children with hearing loss [e.g., 2-5]

An early age of implantation, in fact, fosters the recruitment of a vast neuronal network, thus expanding the opportunities for setting up an efficient neural architecture for the acquisition of language. This line of reasoning is supported by measures of P1 latencies, which vary as a function of chronological age and therefore provide a measure of the central auditory pathway maturation. A study by Sharma et al. [39] showed that “children with CI who had the longest period of auditory deprivation before implantation demonstrated abnormally long cortical response latencies to speech stimuli. On the contrary, those children who had the shortest period of auditory deprivation (3.5 years or less) demonstrated age appropriate latency responses” (p. 511). A subsequent study by Sharma et al. [40] further documented that the development of P1 latencies in two congenitally deaf children, who were implanted at 13 and 14 months of age, respectively, followed a normal trend. In particular, the authors examined the relationship between P1 latencies and the development of canonical babbling; the findings revealed that the development of P1 response latencies and the development of early communicative behaviour followed a similar trajectory. The authors aptly observed that these early stages of speech development in early implanted children “may be positively influenced by the rate of plastic changes in central auditory pathways” [40] (p.511). In other words, early cochlear implantation appears to be optimal for the development of neural correlates of auditory perception because it allows the children to exploit the development of neuronal connections in the brain.

Yet, as many studies observed [e.g., 6-11], early cochlear implantation does not necessarily correspond to age-appropriate language skills. It is important to remain aware that CI may not be sufficient to overcome the risk of weaknesses in several aspects of language [41].

Other factors, such as age at first amplification, type of speech therapy, early sign language exposure, maternal educational level and language input, appear to strongly influence the language development of these children [1]. It should be noted that the Case 1 and 2, relative to Case 3, received the first amplification and therapy a few months later; this condition may have negatively influence their language acquisitions. By contrast, the mothers of the Case 1 and 2 showed higher educational levels than mother of the Case 3. It is widely documented that the maternal educational level plays an important role on child language acquisition in deaf children with CI [42]. Thus, we can exclude that the linguistic discrepancies between Cases 1-2 and the Case 3 are attributable to maternal educational level. There is also recent research that shows that deaf children of hearing parents demonstrate age-level verbal vocabulary growth when exposed to American Sign Language (ASL) by 6 months of age [43]; the lack of access to sign language that characterized the children of the present study may has been a further negative factor for their language development. However, it is important to note that this negative condition affected all three children and not just the first two cases. Thus, taking into account all these conditions and the fact that Cases 1 and 2 showed severely impaired morpho-syntactic abilities whereas the Case 3 age-appropriate skills, it is not unreasonable to put forward the hypothesis that the morpho-syntactic deficit that characterized the Case 1 and 2 may reflects not only the lack of exposure to a natural and optimal hearing during the critical perceptual window but also some concomitant specific impairment to the language system.”

Our data suggest that, in some cases, we may not be facing a simple morpho-syntactic 433 delay due to perceptual difficulties or to the duration of hearing exposure, but a qualitative distortion of the process of language acquisition. A more systematic collection of data is clearly required, yet we feel to suggest that a concomitant occurrence of deafness and 436 morpho-syntactic impairment may explain the variability in linguistic proficiency of  many implanted children.  

 The statements above from your discussion do not take into account the large body of research that looks at neurological development of language acquisition in deaf children with full access to language. I wonder about your case 3 and why they are different; what is your hypothesis. I do not find your warrants strong enough to support either your results or discussion.

R: Following these suggestions, we added literature and reflections concerning the positive effects of the full access to language on verbal acquisition in deaf children. We also better explained that the three cases did not receive this full access to language during the perceptual window. Thus, these not optimal conditions might explain the language deficits of Case 1 and 2.  However, the fact that, despite these similar conditions among the three children, two of them showed severely impaired morpho-syntactic abilities whereas the Case 3 normal skills, seem to pave the way for the hypothesis that the lack of full access to language could only partially explain the linguistic impairment of the first two cases. Thus, it is not unreasonable to suppose that the morpho-syntactic deficit that characterized the Case 1 and 2 may reflects some concomitant specific impairment of the language system. We added these considerations in the Discussion (see previous response to Reviewer 2; see Discussion, pp. 15-17). We also formulated hypotheses of why the Case 3 showed normal morpho-syntactic skills relative to Case 1 and 2 (see Discussion, pp. 16-17).

“In support of this hypothesis, is the qualitatively fine-grained analysis of the language disorders that characterized this single case investigation and that allowed to highlight atypical morpho-syntactical errors in Case 1 and 2 quite similar to those made by children with DLD.”

“Children with DLD show that normal hearing per se is not a sufficient condition for language to be acquired at normal rate [50]. There were also studies that found DLD in signing deaf children, supporting the theoretical argument that DLD can be evident regardless of the modality in which is communicated and can affects deaf children [51-52]. For instance, Herman and colleagues [51] looked at narratives in British Sign Language (BSL) and showed that the DLD-diagnosed deaf children performed poorly on verb morphology, and their narratives exhibited less structure and were shorter than those of the TD children. Marshall and colleagues [52] found deaf children with DLD less able to accurately repeat all elements of BSL sentences, including grammatical constructions. These findings seem to indicate that atypicalities in the sign modality show many of the same characteristics as linguistic deficits in hearing children [53].

Our findings appear in line with these data and seem therefore to provide new evidence to the hypothesis that other single case studies have already suggested [28-29, 44], namely that the deafness could in some case be associated with a DLD, as it would seem in the Case 1 and 2. By contrast, the good morpho-syntactic performances observed in the Case 3 may be the result of both the earlier amplification and therapy relative to other two children with CI and, in line with our hypothesis, language areas integrity.”

Without neurological studies and evidence I cannot accept your conclusion. Additionally, you need to compare unsuccessful CI individuals with Deaf children who should develop a sign language but do not prior to making this leap in your hypothesis development.

R: Our hypothesis of a DLD associated to hearing loss in some children with CI is support by other single case studies (e.g., De Stefani et al., 2009; Hawker et al., 2008) and also by studies on deaf children who should develop a sign language but do not and that, for this reason, received the diagnosis of SLI/DLD (e.g., Herman et al., 2014; Marshall et al., 2015). We cited all these studies in the manuscript (see Discussion, pp. 16-17). However, we recognize the need to replicate the study comparing these two conditions; therefore, we added this need among the limitations/future directions of our study (see Discussion, p. 17). We agree with the Reviewer 2 that our study lacks of neurological evidence or neurofunctional correlates, which are crucial if we aim to explain the association between deafness and language system functional deficit; therefore, despite the results of this single case investigation appear sufficient to suggest this association, we added this issue among the limitations of our study (see Discussion, p. 17). According to this suggestion of the Reviewer 2 and also with a comment of the Reviewer 3 (see below), we also remove the word “damage” in the manuscript, as it may sound like a physical thing, a damage to neural structures of language.

“Children with DLD show that normal hearing per se is not a sufficient condition for language to be acquired at normal rate [50]. There were also studies that found DLD in signing deaf children, supporting the theoretical argument that DLD can be evident regardless of the modality in which is communicated and can affects deaf children [51-52]. For instance, Herman and colleagues [51] looked at narratives in British Sign Language (BSL) and showed that the DLD-diagnosed deaf children performed poorly on verb morphology, and their narratives exhibited less structure and were shorter than those of the TD children. Marshall and colleagues [52] found deaf children with DLD less able to accurately repeat all elements of BSL sentences, including grammatical constructions. These findings seem to indicate that atypicalities in the sign modality show many of the same characteristics as linguistic deficits in hearing children [53].

Our findings appear in line with these data and seem therefore to provide new evidence to the hypothesis that other single case studies have already suggested [28-29, 44], namely that the deafness could in some case be associated with a DLD, as it would seem in the Case 1 and 2. By contrast, the good morpho-syntactic performances observed in the Case 3 may be the result of both the earlier amplification and therapy relative to other two children with CI and, in line with our hypothesis, language areas integrity.”

“Limitations of this work should be acknowledged. First, being a single case study, the generalizability of our findings should be carefully considered. Replication of the present findings with larger samples is clearly needed in the future. In particular, it appears necessary to conduct a detailed qualitative analysis of the language skills in a larger group of deaf children with an earlier age at CI surgery and comparing them with both a group of hearing children with DLD and signing deaf children with DLD. Second, the present work did not consider the role of domain-general neurocognitive processes, such as sequential processing, working memory, executive functions that several authors found to strongly influence the syntactic abilities of children with DLD [54-56] and also the linguistic outcomes of deaf children with CI [18-20]. Third, although we controlled for maternal educational level, we did not investigate other important environmental factors, such as level of parental IQ and the maternal language input, that several studies have shown to explain a considerably proportion of the variance in language acquisition [42]. In effect, it would have been important to check whether Case 1 and 2 received the same kind of language exposure than Case 3 at the early age. Future research should move in these directions. Finally, further limitation of this study is the absence of neurofunctional correlates, which are crucial if we aim to explain the association between deafness and language system functional deficit. Despite these limitations, the results of this single case investigation appear sufficient to suggest this association.”

In terms of language there are a few issues but mostly the English is accurate.

R: We revised the manuscript and corrected some few errors or typo.

Reviewer 3 Report

Hearing thresholds are not perfect descriptors of hearing ability. They are poor descriptors of speech perception in real-life situations. There are several instances where you seem to suggest that restoring audibility should restore language. What children with CI's hear are very different than what normally-hearing kids do. There are some acoustic contrasts for example which might be impossible for them to hear. This makes the acquisition of language different than for normal kids. As someone with absolutely no knowledge of the acoustics of Italian, I am unable to provide examples relevant to this study. Do you think that this might play a role in language acquisition?

Were appropriate consents obtained before the study was performed? Were they compensated in any way? Was the study protocol approved by some appropriate human subjects research board of some sort? Though this seems minor this is an important thing to mention.

Most of your discussion is focused on age of implantation. What do you think the roles of age at first amplification and initiation of speech therapy are? Case 3 received both amplification and therapy much earlier than the other two children with CIs. Also there are factors like language environment at that might interact with these two factors? Is it possible that since both case 1 and 2 did not receive the same kind of language exposure at the early age?

Though mentioned at the end in passing, accounting for extra-linguistic cognitive skills is very important. Top-down processes and cognition might play a vital role in communication in these kids.

Detailed Comments

P1. L18-19. Suggest replacing "highly intelligent" with a more quantitative descriptor that shows how they performed on whatever measurement was used to determine this.

P1. L31. Damage sounds like a physical thing. Are you suggesting there is actual damage to neural structures?

P2. L41. I am not sure I would agree that CI's provide "optimal" conditions for language development.

P2. L43. Please reword "had significant open-set speech recognition at the time of the last postoperative evaluation." This is present in the conclusions of the reference. Please explain that there was a significant improvement post implantation.

P2. L48. Consider replacing "implant" with "implantation"

P2. L57. Consider replacing "performing a CI" with "implanting a CI"

P2. L 71-73. This statement is unclear. Can you explain how references 12 and 13 support this statement especially the bit about the 12 months part of it?

P3. L92-97. Consider omitting the word "instead", breaking them into two sentences. sentence 1: they found a differences between CI kids and normally-hearing kids. Sentence 2: However this difference disappeared when accounting for hearing experience.

P3. L99. Did you mean to say CI compared to normally-hearing children?

P3. 102. Using the word "normal" seems to be vague. Consider a more quantitative approach and not all kids with CIs have "normal" hearing thresholds.

P3. L104. How do you think the age of implantation might be related to length of auditory exposure? Etiology might be an additional factor, hearing aid fitting might make some sounds audible etc.

P3. L109. "Normal"

P3. L124. "Damage" sounds like actual physical changes to neural structures.

P3. L 129. Did you mean "positively"?

P6. Consider presenting the measures before procedure so you do not have to repeat the content in lines 193-203

P6. How long did they testing take for the 3 kids with CIs?

P6. L236. and P7. 264. you say blinded examiners transcribed and scored them and then say one of the authors also scored it? Is this accurate?

Table 2. It might be worth repeating somewhere in the caption that these are error scores. I expected to see raw performance scores and had to re-read the text to understand the table.

P12. L418-... Please discuss age of hearing aid fitting and start of speech therapy. Case 3 was significantly advantaged to receive both these early enough.

P12. L432. "CI"

P12. L 434. It is not clear what "qualitative distortion of the process of language acquisition" means.

P12. L 440. Case3 only had one abnormal error rate compared to the normal hearing kids.

P12. L453. Did you mean "competencies"?

P12. 457-458. What is the support for this idea of "Neuronal structural damage"? Is this the first time that this is coming up? Are there actual studies that show this structural damage? Reference 19 also proposes that their results could arise from damage but does not support it in any way. Plasticity could also explain these results where sensory deprivation could lead to impaired performance which might not be restored to "normal" upon exposure to auditory language.

P13. L482-486. This is unclear. Can you please break it into 2-3 sentences so it is easier to read. Restructure the sentences might also clarify your intent.

The depiction of CIs as a perfect fix to hearing, communication, and language issues throughout the paper is misleading. CIs are not perfect. The assumption that CIs are expected to fix language issues is also overstated. We do not expect CIs to fix communication problems completely in every patient fitted with a CI, so the expectation of further upstream benefits are also equally tempered. Most of the papers on the benefits of CIs, even the ones which show positive effects, usually temper their generalizability and clinical potential. It would serve the audience better if you tempered how you present this expectation. It would be unfair to lead readers, particularly speech therapists and parents, to believe that CIs should fix language problems. Please consider revising the appropriate parts to reflect this concern.

Author Response

R: We thank the Reviewers  for the interest in our study, for the time spent on reviewing our manuscript and for the precious comments and suggestions.

Comments and Suggestions for Authors

Hearing thresholds are not perfect descriptors of hearing ability. They are poor descriptors of speech perception in real-life situations. There are several instances where you seem to suggest that restoring audibility should restore language. What children with CI's hear are very different than what normally-hearing kids do. There are some acoustic contrasts for example which might be impossible for them to hear. This makes the acquisition of language different than for normal kids. As someone with absolutely no knowledge of the acoustics of Italian, I am unable to provide examples relevant to this study. Do you think that this might play a role in language acquisition?

R: We thank the Reviewer 3 for this comment. We agree with the Reviewer 3 that the CIs, while generally effective in transducing sounds into electrical signals for the brain, remain artificial devices that can only approximate the natural hearing experience. Following the suggestion of the Reviewer 3, we modified the manuscript in this direction, highlighting in several instances that this “different hearing” relative to normally-hearing children might play a role in language acquisition (see Introduction, p. 3; see Discussion, p. 14-15).  

“These conflicting effects of the CI on verbal language development have created a significant debate in scientific world, with some researchers that claim that a very early implantation associated to early oral therapy (without exposure to signs) may promote good language performances in children with CI [e.g., 12-13], whereas other argue that CI and oral therapy alone may not be sufficient to ensure typical language development [e.g., 1, 8, 14]. In support of this latter view, there is now evidence that environmental factors (e.g., linguistic input, communication modalities used during interactions, exposure to a sign language) explain a considerably larger proportion of the variance in receptive and productive language than age at implantation [e.g., 15-17]. For example, research has provided initial support to the hypothesis that deaf children with CI can develop better spoken language skills when exposed to a sign language [15-16], demonstrating that early exposure to sign language in a bilingual environment allow the child with CI to express ideas and concepts that he/she was not yet able to speak, thus promoting the development of the language system [16]. Also, domain-general neurocognitive processes, such as sequential processing, working memory, executive functions, may support the language development of these children, or on the contrary hinder it when impaired [e.g., 18-20].

It is clear that understanding variability in CI outcomes requires a broader perspective that goes beyond hearing alone [1]. Also, it is undeniable that, while generally effective in transducing sounds into electrical signals for the brain, CIs remain artificial devices that can only approximate the natural hearing experience. Certainly, the lack of full access to a complete language (verbal and sign language) during the critical perceptual window [21-22] may negatively impact on the neurological development of the language areas [22]. Thus, CIs do not necessary protect the child from failures in full language acquisition [14].”

“In other words, early cochlear implantation appears to be optimal for the development of neural correlates of auditory perception because it allows the children to exploit the development of neuronal connections in the brain.

Yet, as many studies observed [e.g., 6-11], early cochlear implantation does not necessarily correspond to age-appropriate language skills. It is important to remain aware that CI may not be sufficient to overcome the risk of weaknesses in several aspects of language [41].

Other factors, such as age at first amplification, type of speech therapy, early sign language exposure, maternal educational level and language input, appear to strongly influence the language development of these children [1].”

Were appropriate consents obtained before the study was performed? Were they compensated in any way? Was the study protocol approved by some appropriate human subjects research board of some sort? Though this seems minor this is an important thing to mention.

R: According to the Reviewer 3, we added this information in the manuscript (see Method, par. Procedure, p. 9). The participants were not compensated.

“The study met ethical guidelines for human subject protections (Declaration of Helsinki). The children and their parents were informed in detail about the aims of the study, the voluntary nature of their participation, and their right to withdraw from the study at any time. The children’s parents gave informed written consent for participation in the study, data analysis, and data publication.”

Most of your discussion is focused on age of implantation. What do you think the roles of age at first amplification and initiation of speech therapy are? Case 3 received both amplification and therapy much earlier than the other two children with CIs. Also there are factors like language environment at that might interact with these two factors? Is it possible that since both case 1 and 2 did not receive the same kind of language exposure at the early age?

R: As reported in the literature, the age at first amplification and initiation of speech therapy may have a central role on language development of these children; following the suggestion of the Reviewer 3, we added some considerations and literature about this in the manuscript (see Discussion, p. 15-17). We also formulated a hypothesis about the factors that may have promoted good language abilities in the Case 3, relative to Case 1 and 2 (see Discussion, p. 17). We agree with the Reviewer 3 that also environmental factors, such as maternal language input, appear to explain a considerably proportion of the variance in the language acquisition of both TD children and children with CI, as several studies have shown. Unfortunately, we did not observe this maternal variable in our study. However, we added some important considerations about this in the manuscript and we added this issue among the limitations/future directions of the study (see Introduction, p. 3; see Discussion, p. 15, 17).

“Other factors, such as age at first amplification, type of speech therapy, early sign language exposure, maternal educational level and language input, appear to strongly influence the language development of these children [1]. It should be noted that the Case 1 and 2, relative to Case 3, received the first amplification and therapy a few months later; this condition may have negatively influence their language acquisitions. By contrast, the mothers of the Case 1 and 2 showed higher educational levels than mother of the Case 3. It is widely documented that the maternal educational level plays an important role on child language acquisition in deaf children with CI [42]. Thus, we can exclude that the linguistic discrepancies between Cases 1-2 and the Case 3 are attributable to maternal educational level. There is also recent research that shows that deaf children of hearing parents demonstrate age-level verbal vocabulary growth when exposed to American Sign Language (ASL) by 6 months of age [43]; the lack of access to sign language that characterized the children of the present study may has been a further negative factor for their language development. However, it is important to note that this negative condition affected all three children and not just the first two cases. Thus, taking into account all these conditions and the fact that Cases 1 and 2 showed severely impaired morpho-syntactic abilities whereas the Case 3 age-appropriate skills, it is not unreasonable to put forward the hypothesis that the morpho-syntactic deficit that characterized the Case 1 and 2 may reflects not only the lack of exposure to a natural and optimal hearing during the critical perceptual window but also some concomitant specific impairment to the language system.

In support of this hypothesis, is the qualitatively fine-grained analysis of the language disorders that characterized this single case investigation and that allowed to highlight atypical morpho-syntactical errors in Case 1 and 2 quite similar to those made by children with DLD.”

“Our findings appear in line with these data and seem therefore to provide new evidence to the hypothesis that other single case studies have already suggested [28-29, 44], namely that the deafness could in some case be associated with a DLD, as it would seem in the Case 1 and 2. By contrast, the good morpho-syntactic performances observed in the Case 3 may be the result of both the earlier amplification and therapy relative to other two children with CI and, in line with our hypothesis, language areas integrity.”

“These conflicting effects of the CI on verbal language development have created a significant debate in scientific world, with some researchers that claim that a very early implantation associated to early oral therapy (without exposure to signs) may promote good language performances in children with CI [e.g., 12-13], whereas other argue that CI and oral therapy alone may not be sufficient to ensure typical language development [e.g., 1, 8, 14]. In support of this latter view, there is now evidence that environmental factors (e.g., linguistic input, communication modalities used during interactions, exposure to a sign language) explain a considerably larger proportion of the variance in receptive and productive language than age at implantation [e.g., 15-17].”

“Limitations of this work should be acknowledged. First, being a single case study, the generalizability of our findings should be carefully considered. Replication of the present findings with larger samples is clearly needed in the future. In particular, it appears necessary to conduct a detailed qualitative analysis of the language skills in a larger group of deaf children with an earlier age at CI surgery and comparing them with both a group of hearing children with DLD and signing deaf children with DLD. Second, the present work did not consider the role of domain-general neurocognitive processes, such as sequential processing, working memory, executive functions that several authors found to strongly influence the syntactic abilities of children with DLD [54-56] and also the linguistic outcomes of deaf children with CI [18-20]. Third, although we controlled for maternal educational level, we did not investigate other important environmental factors, such as level of parental IQ and the maternal language input, that several studies have shown to explain a considerably proportion of the variance in language acquisition [42]. In effect, it would have been important to check whether Case 1 and 2 received the same kind of language exposure than Case 3 at the early age. Future research should move in these directions. Finally, further limitation of this study is the absence of neurofunctional correlates, which are crucial if we aim to explain the association between deafness and language system functional deficit. Despite these limitations, the results of this single case investigation appear sufficient to suggest this association.”

Though mentioned at the end in passing, accounting for extra-linguistic cognitive skills is very important. Top-down processes and cognition might play a vital role in communication in these kids.

R: We agree with the Reviewer 3; we further highlighted this important issue in the Discussion (see Introduction, p. 3; see Discussion, p. 17).

“Also, domain-general neurocognitive processes, such as sequential processing, working memory, executive functions, may support the language development of these children, or on the contrary hinder it when impaired [e.g., 18-20].

It is clear that understanding variability in CI outcomes requires a broader perspective that goes beyond hearing alone [1]. Also, it is undeniable that, while generally effective in transducing sounds into electrical signals for the brain, CIs remain artificial devices that can only approximate the natural hearing experience. Certainly, the lack of full access to a complete language (verbal and sign language) during the critical perceptual window [21-22] may negatively impact on the neurological development of the language areas [22]. Thus, CIs do not necessary protect the child from failures in full language acquisition [14].”

“Second, the present work did not consider the role of domain-general neurocognitive processes, such as sequential processing, working memory, executive functions that several authors found to strongly influence the syntactic abilities of children with DLD [54-56] and also the linguistic outcomes of deaf children with CI [18-20]. Third, although we controlled for maternal educational level, we did not investigate other important environmental factors, such as level of parental IQ and the maternal language input, that several studies have shown to explain a considerably proportion of the variance in language acquisition [42]. In effect, it would have been important to check whether Case 1 and 2 received the same kind of language exposure than Case 3 at the early age. Future research should move in these directions.”

Detailed Comments

P1. L18-19. Suggest replacing "highly intelligent" with a more quantitative descriptor that shows how they performed on whatever measurement was used to determine this.

R: We have replaced “highly intelligent” with “high cognitive level” (see Abstract, p. 1).

P1. L31. Damage sounds like a physical thing. Are you suggesting there is actual damage to neural structures?

R: We recognized that “damage” may sound like a physical thing, whereas a DLD might have to do with a functional deficit not directly observable at the structure level. According to this suggestion of the Reviewer 3, we removed the word “damage” throughout the manuscript. Moreover, we recognized that our study lacks of neurological evidence or neurofunctional correlates, which are crucial if we aim to explain the association between deafness and language system functional deficit; therefore, despite the results of this single case investigation appear sufficient to suggest this association, we added this issue among the limitations of our study (see Discussion, p. 17).

“Finally, further limitation of this study is the absence of neurofunctional correlates, which are crucial if we aim to explain the association between deafness and language system functional deficit. Despite these limitations, the results of this single case investigation appear sufficient to suggest this association.”

P2. L41. I am not sure I would agree that CI's provide "optimal" conditions for language development.

R: As mentioned above, we agree with the Reviewer 3 that the CIs, while generally effective in transducing sounds into electrical signals for the brain, remain artificial devices that can only approximate the natural hearing experience. We modified all the manuscript in this direction (see Introduction, pp. 2-4).

P2. L43. Please reword "had significant open-set speech recognition at the time of the last postoperative evaluation." This is present in the conclusions of the reference. Please explain that there was a significant improvement post implantation.

R: According to the previous suggestions of both Reviewer 2 and 3, we modified the Introduction and thus this sentence was removed.

P2. L48. Consider replacing "implant" with "implantation"

P2. L57. Consider replacing "performing a CI" with "implanting a CI"

P2. L 71-73. This statement is unclear. Can you explain how references 12 and 13 support this statement especially the bit about the 12 months part of it?

R: We integrated and modified this section making the description of these studies clearer; we also preferred to delate the Park et al. (2019) citation and to add the results that Wie (2010) found in very early implanted children (see Introduction, p. 3).

“Lund [8] showed that the magnitude of difference in language abilities between children with CI and normally hearing children did not appear related to age of implantation or to duration of implantation, and still other studies found significant differences between children with CI and TD children in many linguistic skills (e.g., semantics, syntax, spoken language, pragmatics) with lower performances in the implanted children [9-10]. For instance, Rinaldi and colleagues [9] selected deaf toddlers that received their CI within the second year of life, experienced binaural stimulation, were not exposed to sign language, had not additional disability and had parents actively involved in their child rehabilitation; but despite this careful selection of the sample, they found linguistic skills within normal limits in less than half of the children. The study conducted by Wie [11] in children that received bilateral CIs between 5 and 18 months of age found that, after 12-48 months with CI, only 57% of them had expressive language skills within the normative range.”

P3. L92-97. Consider omitting the word "instead", breaking them into two sentences. sentence 1: they found a differences between CI kids and normally-hearing kids. Sentence 2: However this difference disappeared when accounting for hearing experience.

R: According to the previous suggestions of Reviewer 2, we modified the Introduction and thus this citation and this sentence were removed.

P3. L99. Did you mean to say CI compared to normally-hearing children?

R: The Hawker and colleagues’ study compared six prelingually deaf children with CI (and with disproportionate morpho-syntactic problems) with other six implanted children (control group) matched for etiology, age at implantation, and hearing experience with CI. We modified the sentence in the manuscript making it clearer (see Introduction, p. 4).

“Hawker and colleagues [29] described six prelingually deaf children with disproportionate morfo-syntactic problems, in both production and comprehension, when they were matched to other six implanted children for etiology, age at implantation, and hearing experience with CI.”

P3. 102. Using the word "normal" seems to be vague. Consider a more quantitative approach and not all kids with CIs have "normal" hearing thresholds.

R: We agree with the Reviewer 3; however, according to the previous suggestions of Reviewer 2, we modified the Introduction and thus this sentence was removed.

P3. L104. How do you think the age of implantation might be related to length of auditory exposure? Etiology might be an additional factor, hearing aid fitting might make some sounds audible etc.

R: We agree with the Reviewer 3 that also other factors should be considered, such as etiology, hearing aids fitting, etc. According to the suggestions of both Reviewer 2 and 3, we modified the Introduction in this direction (see previous responses to Reviewer 3).

P3. L109. "Normal"

R: We have modified the word “normal” throughout the manuscript.

P3. L124. "Damage" sounds like actual physical changes to neural structures.

R: We agree with the Reviewer 3 (see previous response to Reviewer 3). According to this suggestion, we removed the word “damage” throughout the manuscript.

P3. L 129. Did you mean "positively"?

R: We have corrected this word.

P6. Consider presenting the measures before procedure so you do not have to repeat the content in lines 193-203

R: Following the suggestion of the Reviewer 3, we presented the Measures section before Procedure section (see Method, par. Measures and Procedure, pp. 8-9).

P6. How long did they testing take for the 3 kids with CIs?

R: For the 3 children with CI, the tests were carried out according to test recommendations, in live voice and given orally, during a 45-min session. We added this information in the manuscript (see Method, par. Procedure, p. 8).

“The three children with CI were evaluated by their speech therapist at the Otolaryngology and Otoneurosurgery Unit of the University of Parma where the children were followed. Specifically, the deaf girls were examined during the individual assessment conducted in the room in which they received the speech therapy. TCGB, PFLI and CPM tests were carried out in a quiet room and were presented according to test recommendations, in live voice and given orally, during a 45-min session.

The TD children were individually met in a quiet room of the school by a psychologist; after a familiarization phase with the examiner, the TCGB, PFLI and CPM tests were administered during a 45-min session.”

P6. L236. and P7. 264. you say blinded examiners transcribed and scored them and then say one of the authors also scored it? Is this accurate?

R: Two blinded examiners, namely first author and the speech therapist, transcribed and scored the narrative productions of the children. No one else did this. We modified this sentence in the Coding section making it clearer (see Method, par. Coding, p. 9).

“With regard to the children’ narrative productions (from both the three deaf children and the control group), they were tape-recorded and then transcribed by two blinded examiners (first author and the speech therapist) for scoring and further inspection (see Appendix).”

“Inter-coder agreement between the two examiners was 90%.”

Table 2. It might be worth repeating somewhere in the caption that these are error scores. I expected to see raw performance scores and had to re-read the text to understand the table.

R: We thank the Reviewer 3 for this suggestion. Following this suggestion, we indicated that the raw scores are error raw scores in both the caption and inside Table 2 (see Table 2).

P12. L418-... Please discuss age of hearing aid fitting and start of speech therapy. Case 3 was significantly advantaged to receive both these early enough.

R: We thank the Reviewer 3 for this suggestion. We agree with the Reviewer 3 that the Case 3 might has been advantaged to receive hearing aid and speech therapy early enough. We added some considerations regarding this in the Discussion (see Discussion, pp. 15-17).

“Other factors, such as age at first amplification, type of speech therapy, early sign language exposure, maternal educational level and language input, appear to strongly influence the language development of these children [1]. It should be noted that the Case 1 and 2, relative to Case 3, received the first amplification and therapy a few months later; this condition may have negatively influence their language acquisitions. By contrast, the mothers of the Case 1 and 2 showed higher educational levels than mother of the Case 3. It is widely documented that the maternal educational level plays an important role on child language acquisition in deaf children with CI [42]. Thus, we can exclude that the linguistic discrepancies between Cases 1-2 and the Case 3 are attributable to maternal educational level. There is also recent research that shows that deaf children of hearing parents demonstrate age-level verbal vocabulary growth when exposed to American Sign Language (ASL) by 6 months of age [43]; the lack of access to sign language that characterized the children of the present study may has been a further negative factor for their language development. However, it is important to note that this negative condition affected all three children and not just the first two cases. Thus, taking into account all these conditions and the fact that Cases 1 and 2 showed severely impaired morpho-syntactic abilities whereas the Case 3 age-appropriate skills, it is not unreasonable to put forward the hypothesis that the morpho-syntactic deficit that characterized the Case 1 and 2 may reflects not only the lack of exposure to a natural and optimal hearing during the critical perceptual window but also some concomitant specific impairment to the language system.

In support of this hypothesis, is the qualitatively fine-grained analysis of the language disorders that characterized this single case investigation and that allowed to highlight atypical morpho-syntactical errors in Case 1 and 2 quite similar to those made by children with DLD.”

“Our findings appear in line with these data and seem therefore to provide new evidence to the hypothesis that other single case studies have already suggested [28-29, 44], namely that the deafness could in some case be associated with a DLD, as it would seem in the Case 1 and 2. By contrast, the good morpho-syntactic performances observed in the Case 3 may be the result of both the earlier amplification and therapy relative to other two children with CI and, in line with our hypothesis, language areas integrity.”

P12. L432. "CI"

R: We have corrected this error.

P12. L 434. It is not clear what "qualitative distortion of the process of language acquisition" means.

R: We modified the sentence making it clearer (see Discussion, p. 16).

“Our data suggest that, in some cases, we may not be facing a simple morpho-syntactic delay due to perceptual difficulties or to the duration of hearing exposure, but a disorder of the processes of language acquisition. A more systematic collection of data is clearly required, yet we feel to suggest that a concomitant occurrence of deafness and morpho-syntactic impairment may explain a proportion of variability in linguistic proficiency of some implanted children.”

P12. L 440. Case3 only had one abnormal error rate compared to the normal hearing kids.

R: We added this consideration in the Discussion (see Discussion, p. 16).

“Since the discrepancies between Case 1-2 and Case 3 cut across the entire set of syntactic structures investigated by the TCGB except one (Case 3 only had one abnormal error rate on passive affirmative sentences, compared to the normative range), it appears that the grammatical errors made by the first two deaf children may reflect an extended impairment to the morpho-syntactic system.”

P12. L453. Did you mean "competencies"?

R: We have corrected this error.

P12. 457-458. What is the support for this idea of "Neuronal structural damage"? Is this the first time that this is coming up? Are there actual studies that show this structural damage? Reference 19 also proposes that their results could arise from damage but does not support it in any way. Plasticity could also explain these results where sensory deprivation could lead to impaired performance which might not be restored to "normal" upon exposure to auditory language.

R: As reported above, we followed the suggestion of the Reviewer 3, removing the words “damage” and “neuronal structural damage” throughout the manuscript and making our hypothesis clearer (see previous response to Reviewer 3; see Discussion, pp. 16-17; see Conclusions, pp. 18-19).

“Children with DLD show that normal hearing per se is not a sufficient condition for language to be acquired at normal rate [50]. There were also studies that found DLD in signing deaf children, supporting the theoretical argument that DLD can be evident regardless of the modality in which is communicated and can affects deaf children [51-52]. For instance, Herman and colleagues [51] looked at narratives in British Sign Language (BSL) and showed that the DLD-diagnosed deaf children performed poorly on verb morphology, and their narratives exhibited less structure and were shorter than those of the TD children. Marshall and colleagues [52] found deaf children with DLD less able to accurately repeat all elements of BSL sentences, including grammatical constructions. These findings seem to indicate that atypicalities in the sign modality show many of the same characteristics as linguistic deficits in hearing children [53].

Our findings appear in line with these data and seem therefore to provide new evidence to the hypothesis that other single case studies have already suggested [28-29, 44], namely that the deafness could in some case be associated with a DLD, as it would seem in the Case 1 and 2. By contrast, the good morpho-syntactic performances observed in the Case 3 may be the result of both the earlier amplification and therapy relative to other two children with CI and, in line with our hypothesis, language areas integrity.”

“Numerous variables, such as age at first amplification and at beginning language rehabilitation, early sign language exposure, age at implantation, cognitive skills, and maternal education level and language input, may influence the linguistic outcomes of deaf children with CI; as previously reported in this work, these variables have been extensively studied in this clinical population. Instead, few are the studies that investigated if the reported delay in language acquisition among implanted children may be characterized as a domain specific language disorder.Case 1 and 2  showed grammar impairments very similar to those of children with DLD. Thus, the profound morpho-syntactic deficits observed in these two cases may be the result of more co-occurring factors: deafness-related factors, such as the lack of full access to a complete language (verbal and sign language) during the critical perceptual window [1, 21-22], and also the presence of a concomitant DLD [28-29, 44].

In other words, we are suggesting that the persistent language impairment in some deaf children with CI may be the final outcome of a pre-existing impairment to the language system, combined with the negative effects of a profound congenital deafness.”

P13. L482-486. This is unclear. Can you please break it into 2-3 sentences so it is easier to read. Restructure the sentences might also clarify your intent.

R: We have restructured the sentences (see Conclusion, p. 18).

“Early identification of domain specific language impairments in these children may allow for planning and implementation of individualized interventions. To this end, a qualitative and accurate analysis of the type of morpho-syntactic errors appears needed during the language assessment of these children. Also, interventions are needed that specifically act on affected language domains, similarly to what is done with children with DLD.”

The depiction of CIs as a perfect fix to hearing, communication, and language issues throughout the paper is misleading. CIs are not perfect. The assumption that CIs are expected to fix language issues is also overstated. We do not expect CIs to fix communication problems completely in every patient fitted with a CI, so the expectation of further upstream benefits are also equally tempered. Most of the papers on the benefits of CIs, even the ones which show positive effects, usually temper their generalizability and clinical potential. It would serve the audience better if you tempered how you present this expectation. It would be unfair to lead readers, particularly speech therapists and parents, to believe that CIs should fix language problems. Please consider revising the appropriate parts to reflect this concern.

R: We thank the Reviewer 3 for this important reflection. We agree with the Reviewer 3 and we too think that it is really unfair to lead readers, particularly speech therapists and parents, to believe that CIs should fix language problems. We modified the manuscript following this suggestion.  

Round 2

Reviewer 2 Report

I note that you have taken into account many of my earlier recommendations. Thank you for that.  here are my additional comments

Behavioral Sciences

Behavsci-1267580

Morpho-syntactic impairment

The first sentence in the abstract has hystory which my computer just changed, and I had to change back.  Also, you have two periods. I am not going to note these types of issues again, but please review the paper for these kinds of issues. Many sentences are awkward in their English. Additionally, one sentence paragraphs are not paragraphs

I take issue with your first sentence in the Introduction.  Implants do not provide any natural contexts and incidental learning is still limited – frequently to what is in the visual field to allow lip reading – rather than the natural experience of hearing people “hearing behind the head.”

paragraphs are repeating themselves—see lines 60-71 is the same as 43 to 48

As noted by all of your citations the most effective number I have found by page three is 57% of CI deaf children having skills in the normative range---even if it is the lower range. Pat Kuhl and others show that the babbling period of 6 months to 10/11 months is when these skills appear to set up—CIs miss that period.  Therefore, with three children I do not accept your hypothesis that these children are showing Developmental Delay as would a normal hearing child. There are simply TOO many non-controlled factors such as parent income, parental involvement etc to make this hypothesis

Your section on the impact of adding sign language to the cognitive context then provides the deaf infant the perceptual window for neurolinguistic development ---my hypothesis is that this is the key – not that there is another deficit

You have added some of this information. There IS research on deaf children with deaf parents who do have Specific Language Impairment---however adding in the CI I believe complicates this issue and provides so many confounds that it will be difficult to determine that hypothesis.  Newer technologies of imaging may help in the future but only a sample of three here is not convincing for me